# Cold-to-warm flow regime transition in snow avalanches

Köhler Anselm[1,2], Fischer Jan-Thomas[2], Scandroglio Riccardo[1], Bavay Mathias[1], McElwaine Jim[3], and Sovilla Betty[1]

[1]WSL Institute for Snow and Avalanche Research SLF, Davos Dorf, Switzerland
[2]Austrian Research Centre for Forests (BFW), Innsbruck, Austria
[3]Department of Earth Sciences, Durham University, Durham, UK

*Correspondence to:* Anselm Köhler, anselm.koehler@slf.ch

**Abstract.** Large avalanches usually encounter different snow conditions along their track. When they release as slab avalanches comprising cold snow, they can subsequently develop into powder snow avalanches entraining snow as they move down the mountain. Typically, this entrained snow will be cold ($\overline{T} < -1\,°\mathrm{C}$) at high elevations near the surface, but warm ($\overline{T} > -1\,°\mathrm{C}$) at lower elevations or deeper in the snow pack. The intake of warm snow is believed to be of major importance to increase the temperature of the snow composition in the avalanche and eventually cause a flow regime transition. Measurements of flow regime transitions are performed at the Vallée de la Sionne avalanche test site in Switzerland using two different radar systems. The data are then combined with snow temperatures calculated with the snow cover model `SNOWPACK`. We define transitions as *complete*, when the deposit at runout is characterized only by warm snow, or as *partial*, if there is a warm flow regime but the farthest deposit is characterized by cold snow. We introduce a transition index $F_t$, based on the runout of cold and warm flow regimes, as a measure to quantify the transition type. Finally, we parameterize the snow cover temperature along the avalanche track by the altitude $H_s$, which represents the point where the average temperature of the uppermost $0.5\,\mathrm{m}$ changes from cold to warm. We find that $F_t$ is related to the snow cover properties, *i.e.* approximately proportional to $H_s$. Thus, the flow regime in the runout area and the type of transition can be predicted by knowing the snow cover temperature distribution. We find, that, if $H_s$ is more than $500\,\mathrm{m}$ above the valley floor for the path geometry of Vallée de la Sionne, entrainment of warm surface snow leads to a complete flow regime transition and the runout area is reached by only warm flow regimes. Such knowledge is of great importance since the impact pressure and the effectiveness of protection measures are greatly dependent on the flow regime.

## 1 Introduction

For avalanche practitioners dealing with situations where they need to judge the avalanche hazard for infrastructure, flow regime transitions can cause large uncertainties. Which flow regime reaches the valley bottom is of great interest from two perspectives. Firstly, the usefulness of permanent protection measures like avalanche dams depends strongly on the flow regime (Jóhannesson et al., 2009). Indeed, deflecting and catching dams are relatively ineffective against the highly fluidized intermittent frontal regime of powder snow avalanches, whereas dense flow regimes, especially warm regimes, can more easily be diverted or even stopped. Secondly, the force generated by an avalanche on a structure in the path depends strongly on flow regime (Gauer

et al., 2008b). A velocity dependent grain-inertia induced pressure is dominant in cold-dry flow regimes, whereas a flow depth dependent quasi-static gravitational contribution is dominant in warm-wet flow regimes (Sovilla et al., 2016).

Recent studies identified the snow temperature as a key parameter causing the agglomeration of snow (Steinkogler et al., 2014) and a change of the flow dynamics by altering the velocity and the effective friction (Naaim et al., 2013; Gauer and Kristensen, 2016), as well as the stopping dynamics (Köhler et al., 2018). A temperature value of $-1\,°C$ is proposed by a study on snow granulation (Steinkogler et al., 2015a), where they observed a significant change from millimetre sized grains to the formation of decimetre sized granules above this temperature. We emphasize the temperature of the snow by calling avalanches warm and cold rather than wet and dry, since the flow behaviour changes already at a threshold of $-1\,°C$. That this transition occurs below $0\,°C$ is presumably due to the existence of a quasi-liquid layer even at sub-zero temperatures (Dash et al., 2006; Turnbull, 2011). Liquid water may cause the cohesion of snow to increase by formation of granules, but may also lubricate the contacts between snow aggregates and result in slush flows.

The avalanche flow regime – a region inside the avalanche where the same physical processes are dominant – can be deduced from radar signatures of flow processes by use of the radar GEODAR (Köhler et al., 2018). Cold flow regimes are identified by the starving mechanism in which the avalanche loses mass from the tail until finally the front comes to halt. We call cold regimes those flow regimes which contain cold snow ($< -1\,°C$), and are either the cold dense regime or the dilute frontal region of powder snow avalanches called intermittent regime. In contrast, warm flow regimes are identified by either abrupt stopping or a backward propagating shock; either a large flowing part stops instantaneously or the front comes to a halt and incoming material piles up. We call warm regimes those flow regimes which occur for warm snow temperatures ($> -1\,°C$), and are either the warm shear regime or the warm plug regime. Köhler et al. (2018) differentiated flow regimes comprising cold and warm snow further in detail. However, relevant for the discussion here is that the majority of large avalanches shows transitions between cold and warm flow regimes. These transitions and the relation with snow cover properties are the focus of this paper.

This study deals exclusively with avalanches that start in a cold-dry regime and parts of which untergo a transition to a warm-wet regime, that is, those avalanches exhibit a cold-to-warm flow regime transition. We define these transitions as *partial transition* or *complete transition*, depending on whether only parts, or the entire avalanche, transforms. A partial transition becomes often visible at the tail of the flowing avalanche as cold and warm flow regimes separate due to different velocities and the final runout is still cold-dominated. With a complete transition, all the snow becomes warm and the final runout is determined by the dynamical properties of the warm flow regime.

Large avalanches composed mostly of cold snow are powder snow avalanches and have been described by many authors (Sovilla et al., 2015; Issler, 2003). They usually release as a slab containing cold snow, and the runout area is reached by fast flowing cold snow. In addition to the typical structure of a suspension cloud, a frontal intermittent region and a cold dense core and tail, GEODAR images reveal often warm flow regimes in the tail and indicate that a partial transition happened (Köhler et al., 2018). Issler (2003) introduced the nomenclature "mixed powder snow avalanche" to describe the occurrence of dilute and dense flow regimes together in one avalanche event. The definition applies mostly for cold dense and dilute regimes, but

Issler (2003) reported of damp deposits which are not covered by dust of the dilute regimes and thus had been flowing later and more slowly.

Warm-dominated avalanches release similarly to cold-dominated ones, but transform completely somewhere along the path. In this case, the runout is dominated by warm regimes. Literature on this type of avalanche is hard to find, since to our knowledge such a transition is rarely recognized and the events are rather described as wet avalanches. There are some measurements with radar and picture in Gauer et al. (2008a) indicating a complete transition, but have been interpreted as a secondary wet slab released by the primary dry–cold avalanche. An example of an avalanche with a complete transition released spontaneously near the village of Moos in Passeiertal, Italy, on the 6th of February 2014. A video of this avalanche drew a lot of attention, because most of the avalanche travelled along a road in front of houses with people on their balconies (www.youtube.com/watch?v=f5waSw2mMfY). The avalanche released on the south-east facing slopes below the summit of Scheibkopf (2816 m a.s.l.) after a major snow storm and developed a large powder cloud and thus contained cold snow. At around 15 s after the start of the video, the powder cloud began to decay so that the cold parts stopped at approximately 1700 m a.s.l.. A dense flow continued and flowed over a cliff into a shallow valley, where finally a slow-moving plug flow developed. The avalanche transformed completely from a cold powder snow avalanche into a warm flow, which finally flowed slowly along the road.

The present study tries to answer the question of how the *degree* of transition relates to the snow cover properties along the avalanche track. To quantitatively describe the *degree* of transition as a continuum between partial and complete, we define the transition index $F_t$, which is a function of path length of warm and cold flow regimes.

We then explore the relationship with snow cover characteristics, focusing on the snow temperature $\overline{T}$ averaged over the uppermost $0.5$ m of the snow cover. This depth is expected to be frequently entrained into the avalanche, though of course there may be more or less entrainment. This assumption is backed up by field observations on typical entrainment depths and underpinned in section 2.2. We find that $\overline{T}$ is a representative indicator for the thermal energy intake due to entrainment and we will show that it can be used to give a good prediction of the transition index.

The study starts by introducing the test site and sensor equipment (sec. 2.1), the method to derive the snow cover temperatures by simulations with the numerical model `SNOWPACK` (sec. 2.2), and a short description of the avalanche data (sec. 2.3). The following results section is divided in two parts. Firstly, we detail the kinematic and dynamic characteristics of partial and complete transitions by means of two different radar systems (sec. 3.1 and 3.2). Secondly, we present the analysis of the degree of transition with the snow cover temperatures (sec. 3.3). Finally, the discussion (sec. 4) is divided into two sections which brings results into wider context and points out limitations of our methodology.

## 2 Methods and data

### 2.1 Test site and radar sensors

The full-scale avalanche test site Vallée de la Sionne (VdlS) is situated in the west of Switzerland. The east-facing avalanche path extends from high altitudes at $2700$ m a.s.l. to intermediate altitudes with a total drop-height of $1300$ m. The VdlS

avalanche track can be roughly characterized with a 40° steep release area above 2300 m a.s.l., followed by a flatter section which leads into two 35° steep couloirs between 1800 m a.s.l. and 2100 m a.s.l. with the runout area starting below and continuing into the valley floor at 1400 m a.s.l.. Especially in the early and late season, there can be minimal snow in the lower part of the slope but still sufficient snow for avalanches in the release areas at higher elevations.

The test site is equipped with multiple sensor systems at different locations. On a 20 m high pylon near the start of the runout area, sensors give high-resolution vertical profiles of flow velocity, flow height, density as well as impact pressure (Sovilla et al., 2013). Upward-looking flow profiling radars and seismic sensors are situated in two locations along the flow path. Data are also collected over the entire slope by two complementary radar systems: the GEODAR (Ash et al., 2010) allows tracking of avalanche features with high spatial and temporal resolution, and the pulse-Doppler system (Schreiber
et al., 2001) complements this with complete velocity distributions of the avalanche flow. An automatic seismic trigger enables measurement of even spontaneous avalanches.

GEODAR is a high-resolution frequency modulated continuous wave radar and was first installed in winter season 2009/10 (Ash et al., 2014). The system has been continually improved and currently has a range resolution of 0.75 m at 110 Hz over the entire slope (Köhler et al., 2018). GEODAR is able to resolve internal flow structures below the powder cloud. By means
of feature tracking, comparison with other data and qualitative interpretations, new and very detailed insights into processes during an avalanche descent have been gained (Vriend et al., 2013; Köhler et al., 2016, 2018). The data processing, feature extraction and terrain registration are done here with the same methods as described in these three publications. An approach velocity $v_a(t)$ of the avalanche front towards the radar is calculated by the derivative of the range-time trajectory $r(t)$, which is corrected for the angle between terrain and the radar beam $\theta$ (Köhler et al., 2016)

$$v_a(t) = \frac{\dot{r}(t)}{\cos\theta} \,. \tag{1}$$

The processed GEODAR data are usually shown as range-time plots with the colour representing the intensity of the moving-target identification (MTI) filter (*e.g.* left panels of Fig. 2). This filter suppresses static targets and background clutter and highlights moving structures. Often the front and tail give the clearest signature from light to dark colours and vice versa. Inbetween, the avalanche signature is usually dark coloured with line and streak patterns (Köhler et al., 2018). The distance
between front and the tail along the range axis is the avalanches flowing length, which in general increases between release area and the fastest parts reaching the valley floor.

The other radar, a pulse-Doppler radar, was permanently installed at Vallée de la Sionne for the winter season 2009/10 and upgraded in 2016/17. The older system provided a spatial resolution of $R_g = 50$ m (Schreiber et al., 2001) and the newer system gives $R_g = 25$ m (Fischer et al., 2016). This resolution is referred to a range gate extent ($R_g$), and the Doppler measurements
provide an intensity distribution of velocities over time $I_k(t,v)$ of the flowing material within each range gate $R_k$ with a running number $k$ (*e.g.* Fig. 3 and 4). The peak of this distribution describes the velocity of maximum intensity and gives the velocity at which most of the material is travelling (Gauer et al., 2007; Fischer et al., 2014). The data can also be transformed into a range-time representation (Fischer et al., 2016), which is very similar to GEODAR intensity-range plots but represents

the mean velocity in each range gate $k$ at each time as

$$\overline{v}_k(t) = \frac{\int v \, I_k(t,v) \, dv}{\int I_k(t,v) \, dv} \qquad (2)$$

(middle panels in Figure 2). This can then be converted from a discrete function of range to a continuous function using finite volume interpolation methods.

## 2.2 Snow cover reconstruction

The test site Vallée de la Sionne is equipped with three weather stations. The bottom station VDS3 (indicated with subscript $_3$) at elevation $H_3 = 1680 \, \mathrm{m}$ a.s.l. is representative for the runout area. The top weather station VDS2 (subscript $_2$) at elevation $H_2 = 2390 \, \mathrm{m}$ a.s.l. gives a good approximation for the release area even though it is situated $3 \, \mathrm{km}$ to the north of the avalanche path. Both weather stations are installed in flat fields sheltered from winds to most-accurately represent the undisturbed snow height. Both weather stations measure air temperature, humidity, wind speed, snow height, radiation and snow surface temperature, which are the complete set of parameters necessary to simulate the desired snow cover profiles. A third station VDS1 is situated directly on the ridge above the release area and measures especially wind speed and therefore wind loading.

The meteorological data have been prepared with the library `meteoIO` (Bavay and Egger, 2014), *i.e.* missing values have been interpolated and temperature and snow height data have been filtered. Corrections according to Huwald et al. (2009) were necessary for the air temperature, as unventilated temperature sensors are used and these usually overestimate the temperature for situations with low wind speed but strong radiation. Special attention has been given also to the snow height data at the VDS3 station, since for low snow heights the measurements were biased by vegetation so that the values had to be manually reset to $0 \, \mathrm{m}$.

To obtain snow temperature profiles, the snow cover has to be modelled as these are not measured automatically. The snow cover at the location of the meteo stations has been reconstructed with the numerical energy balance model `SNOWPACK` (Lehning et al., 2002) to obtain vertical snow profiles as a function of time. We have applied the simulation setup for the operational simulations of the Intercantonal Measurement and Information System (IMIS), the high alpine weather station network in Switzerland (Schmucki et al., 2014).

In this publication, we explore how the temperature of the snow cover entering an avalanche determines the degree of a cold-to-warm transition. There is no common approach to reduce the temperature profile of the snow cover to a single representative value. Naaim et al. (2013) used the average snow temperature in the full path without differentiating between release and runout area. This approach is very broad, but suitable for situations where it is necessary to compare a large number of avalanche events. In a detailed study, Steinkogler et al. (2014) averaged over an estimated entrainment depth. This is most accurate but requires very detailed entrainment data, and therefore is only suited for studies with a few avalanches. Köhler et al. (2018) approximated this depth by assuming that the uppermost $0.5 \, \mathrm{m}$ of snow was entrained. Sovilla et al. (2006) showed that significant entrainment occurs along the full avalanche path. If we divide the typical volume of large avalanches in VdlS of $(0.5\text{--}1) \times 10^6 \, \mathrm{m}^3$ by the typical affected area of $(1\text{--}2) \times 10^6 \, \mathrm{m}^2$ (Dufour et al., 2000; Steinkogler et al., 2014), the average entrainment depth of $\overline{h} = 0.5 \, \mathrm{m}$ appears to be a reasonable assumption. The approach with a constant averaging depth can be

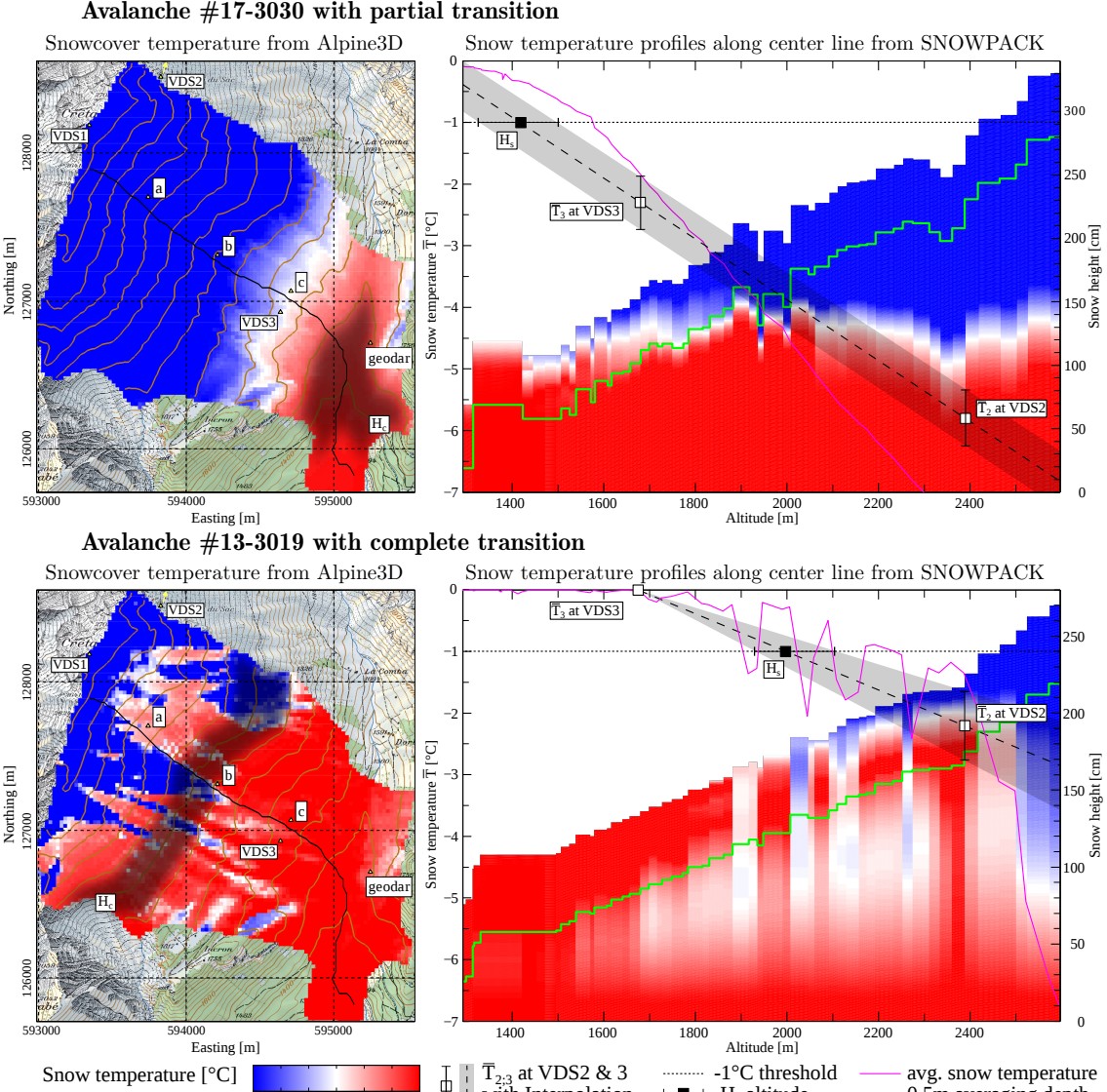

**Figure 1.** Snow cover simulations for avalanche #17-3030 with partial transition (top) and #13-3019 with complete transition (bottom). The left panels show the averaged temperatures $\overline{T}$ of the uppermost $0.5\,\mathrm{m}$ snow cover from `Alpine3D` gridded over the VdlS catchment. The area overlay in grey denotes $H_s \pm \triangle H_s$. For reference, the location of the pylon (c) and profiling radars (a, b) are shown. The right panels show the snow temperature profiles along the path of steepest descent (black line in left panels). The purple curve indicates the average temperature $\overline{T}$ of the top $0.5\,\mathrm{m}$ of each vertical profile. $\overline{T}$ at the top and bottom weather station are shown with white squares together with the temperature variations. $H_s$ is calculated using linear interpolation between the weather stations as the intercept of grey area with $-1\,°\mathrm{C}$ and shown with black squares.

regarded as a trade-off between accuracy and practicability for analysing many avalanche events, even though large avalanches can usually dig much deeper into the snow cover (Gauer and Issler, 2004; Sovilla et al., 2006).

Thus, we average the simulated snow temperature

$$\overline{T} = \sum_i \frac{h_i T_i}{\overline{h}} \tag{3}$$

from the layers $i$ with thickness $h_i$ and layer temperature $T_i$ in the uppermost $\overline{h} = 0.5\,\mathrm{m}$ of the simulated snow cover. With `SNOWPACK` simulations we compute $\overline{T}$ only at the location of VDS2 and VDS3 (squares in right panel of Fig. 1), but are interested in the snow cover temperatures along the entire avalanche path.

We parameterize $\overline{T}$ along the avalanche path with the altitude $H_s$ of the $-1\,°\mathrm{C}$ line. $H_s$ represents the altitude where $\overline{T}$ crosses the threshold from above to below $-1\,°\mathrm{C}$, similar to the zero-degree level in meteorology. Motivated by the work of Steinkogler et al. (2014), we estimate $H_s$ with a linear relation between the altitude of the weather stations $H_2$ and $H_3$ and the average temperatures $\overline{T}_2$ and $\overline{T}_3$ of the uppermost $0.5\,\mathrm{m}$ of the snow cover by

$$H_s = H_3 + (H_2 - H_3)\frac{-1 - \overline{T}_3}{\overline{T}_2 - \overline{T}_3} . \tag{4}$$

The elevation uncertainty $\triangle H_s$ is estimated with the temperature variation of the uppermost $0.5\,\mathrm{m}$ snow temperature $\triangle \overline{T}_{2;3}$ at both weather stations. In fact, $\triangle \overline{T}_{2;3}$ is the standard deviation of the simulated layer temperatures in the uppermost $0.5\,\mathrm{m}$ of the snow pack. The right panels of Figure 1 show graphically the linear interpolation of $\overline{T}_{2;3}$, and $H_s$ and $\triangle H_s$ is found at the intercept of the grey area with the dashed line at temperature $-1\,°\mathrm{C}$. $\triangle H_s$ is not the uncertainty of $H_s$, but give rather a spread of possible values.

Our parameterization of the snow cover temperatures in the avalanche path and the temperature gradient is in fact only dependent on altitude. To check the validity of these strong assumptions (flat field simulations, linear elevation gradient, see Eq. 4), we have additionally performed `Alpine3D` simulations to compare the results (Lehning et al., 2006). `Alpine3D` performs physically-based spatial interpolations of all the meteorological input data over a domain, *i.e.* the area of the VDLS test site. This domain is sliced into grid cells with resolution of $25\,\mathrm{m}$ x $25\,\mathrm{m}$ and for each cell a `SNOWPACK` simulation is performed (Schlögl et al., 2016). While our single `SNOWPACK` simulations are calculated for flat fields, `Alpine3D` simulates the snow cover at each cell with their local slope and aspect. The `Alpine3D` output are grids of a parameter like the $0.5\,\mathrm{m}$ snow temperature $\overline{T}$ for every simulation step, and a full `SNOWPACK` output can be generated at any point of interest.

Results of `Alpine3D` and `SNOWPACK` simulations for two example avalanches are shown in Figure 1. The left panels show the spatial distribution of the temperature $\overline{T}$ over the catchment of VdlS from `Alpine3D`. The right panels display the vertical profiles of layer temperatures $T_i$ along the line of steepest descent from the release area. These profiles are generated by the `Alpine3D` simulations as full outputs at points of interest. Additionally in the right panels are the data of `SNOWPACK` simulations for both weather stations denoted with white squares, together with a graphical representation of the interpolation in Eq. 4 which give $H_s$ at the intercept with the temperature threshold of $-1\,°\mathrm{C}$ (black square).

The two example in Figure 1 have the largest deviation between Eq. 4 and the `Alpine3D` simulations in our data sets. The #17-3030 event (top) occurred in spring-time when the flat fields receive more sun than the eastern aspects and thus show higher

**Table 1.** Summary of the avalanche events with the extracted path lengths $P$, the transition index $F_t = \frac{P_w - P_c}{\max(P_c, P_w)}$ and altitude of transition $H_t$, as well as the snowpack conditions $H_s$ and mean temperatures at both meteo stations $\overline{T}_{2;3}$. Data of avalanche events indicated with a $^*$ in front of the row can be received from the GEODAR repository (McElwaine et al., 2017).

| SLF-Nr | GEODAR timestamp | $P_c$ [m] | $P_w$ [m] | $F_t$ | $H_t$ [m a.s.l.] | $H_s$ [m a.s.l.] | $\overline{T}_2$ [°C] | $\overline{T}_3$ [°C] |
|---|---|---|---|---|---|---|---|---|
| $^*$#13-3003 | 2012-12-04-04-46-05 | 1980 | 1770 | -0.11 | 1820 | $1719 \pm 30$ | $-4.4 \pm 0.2$ | $-0.8 \pm 0.1$ |
| $^*$#13-3019 | 2013-02-01-17-14-50 | 1630 | 2370 | 0.31 | 1730 | $1989 \pm 74$ | $-2.3 \pm 0.6$ | $0.0 \pm 0.0$ |
| $^*$#13-3020 | 2013-02-01-20-18-46 | 1990 | 2580 | 0.23 | 1660 | $2003 \pm 44$ | $-2.2 \pm 0.3$ | $0.0 \pm 0.0$ |
| $^*$#13-3021 | 2013-02-02-05-27-31 | 1560 | 2230 | 0.30 | 1700 | $1953 \pm 26$ | $-2.6 \pm 0.2$ | $0.0 \pm 0.0$ |
| $^*$#13-3024 | 2013-02-05-23-31-53 | 2080 | 1630 | -0.22 | 1770 | $1506 \pm 146$ | $-8.1 \pm 1.1$ | $-2.4 \pm 1.1$ |
| $^*$#14-0012 | 2014-02-13-19-21-32 | 2460 | 1630 | -0.34 | 1770 | $1325 \pm 73$ | $-4.3 \pm 0.6$ | $-2.1 \pm 0.3$ |
| $^*$#15-0009 | 2015-01-29-05-18-08 | 1980 | 1580 | -0.20 | 1810 | $1627 \pm 82$ | $-5.3 \pm 0.3$ | $-1.3 \pm 0.5$ |
| $^*$#15-0013 | 2015-01-30-02-12-22 | 2640 | 1680 | -0.36 | 1810 | $1200 \pm 221$ | $-7.2 \pm 0.4$ | $-3.5 \pm 0.8$ |
| $^*$#15-0016 | 2015-02-03-10-20-16 | 2310 | 1200 | -0.48 | 1870 | $1281 \pm 191$ | $-9.9 \pm 0.8$ | $-4.2 \pm 1.2$ |
| $^*$#15-0020 | 2015-02-03-12-04-39 | 2560 | 1860 | -0.27 | 1770 | $1585 \pm 71$ | $-8.6 \pm 0.5$ | $-1.9 \pm 0.7$ |
| #16-3017 | 2016-01-18-10-40-14 | 2640 | 1370 | -0.48 | 1970 | $1556 \pm 16$ | $-10.4 \pm 1.0$ | $-2.4 \pm 0.3$ |
| #16-3032 | 2016-02-09-18-31-25 | 1430 | 1430 | 0.00 | 1960 | $1858 \pm 50$ | $-3.4 \pm 0.5$ | $-0.2 \pm 0.1$ |
| #17-3014 | 2017-01-13-02-47-38 | 1760 | 1560 | -0.11 | 1790 | $1470 \pm 45$ | $-4.5 \pm 0.3$ | $-1.8 \pm 0.2$ |
| #17-3027 | 2017-03-02-12-22-03 | 1590 | 1820 | 0.13 | 1820 | $1979 \pm 121$ | $-2.1 \pm 0.7$ | $-0.2 \pm 0.1$ |
| #17-3028 | 2017-03-06-15-48-07 | 1990 | 1530 | -0.23 | 1850 | $1798 \pm 72$ | $-4.0 \pm 0.7$ | $-0.4 \pm 0.3$ |
| #17-3030 | 2017-03-06-22-05-22 | 2600 | 2140 | -0.18 | 1750 | $1416 \pm 77$ | $-5.8 \pm 0.5$ | $-2.3 \pm 0.4$ |
| #17-3033 | 2017-03-08-11-04-22 | 2130 | 2130 | 0.00 | 1730 | $1786 \pm 35$ | $-4.4 \pm 0.5$ | $-0.4 \pm 0.1$ |
| #17-3036 | 2017-03-08-11-25-24 | 2090 | 1930 | -0.08 | 1690 | $1786 \pm 35$ | $-4.4 \pm 0.5$ | $-0.4 \pm 0.1$ |
| Moos avalanche, 6 Feb. 2014 | | 1600 | 2900 | 0.45 | 1700 | $> 2000$ | – | – |

temperature for $\overline{T}_2$ at the station VDS2. The event #13-3019 corresponds to a rain event and the right panel shows isothermal $0\,°C$ snow in the runout area but very cold snow in the release area. However, if compared with the gridded $\overline{T}$ of `Alpine3D` output in the left panels, both $H_s$ estimates (grey areas) reflect the pattern of warm and cold temperatures reasonably well. Thus, we expect a deviation from equation 4 for situations like spring-time with strong radiation influence, and $H_s$ will be less

5 accurate if large regions are isothermal. In particular, rain-on-snow events may be overlooked as the water ingress is difficult to measure and to capture with `SNOWPACK` (Würzer et al., 2017).

## 2.3 Data set

In this study, we selected avalanche events from Vallée de la Sionne that fulfill three criteria: 1) they were large enough to pass the measurement pylon at range $655\,m$ near the start of the runout area. This criterion implies a minimum drop height

10 of $1000\,m$. 2) The avalanche stopped where it was visible to GEODAR, that is before the counter-slope. 3) A cold-to-warm transition as described by Köhler et al. (2018) occurred somewhere in the avalanche.

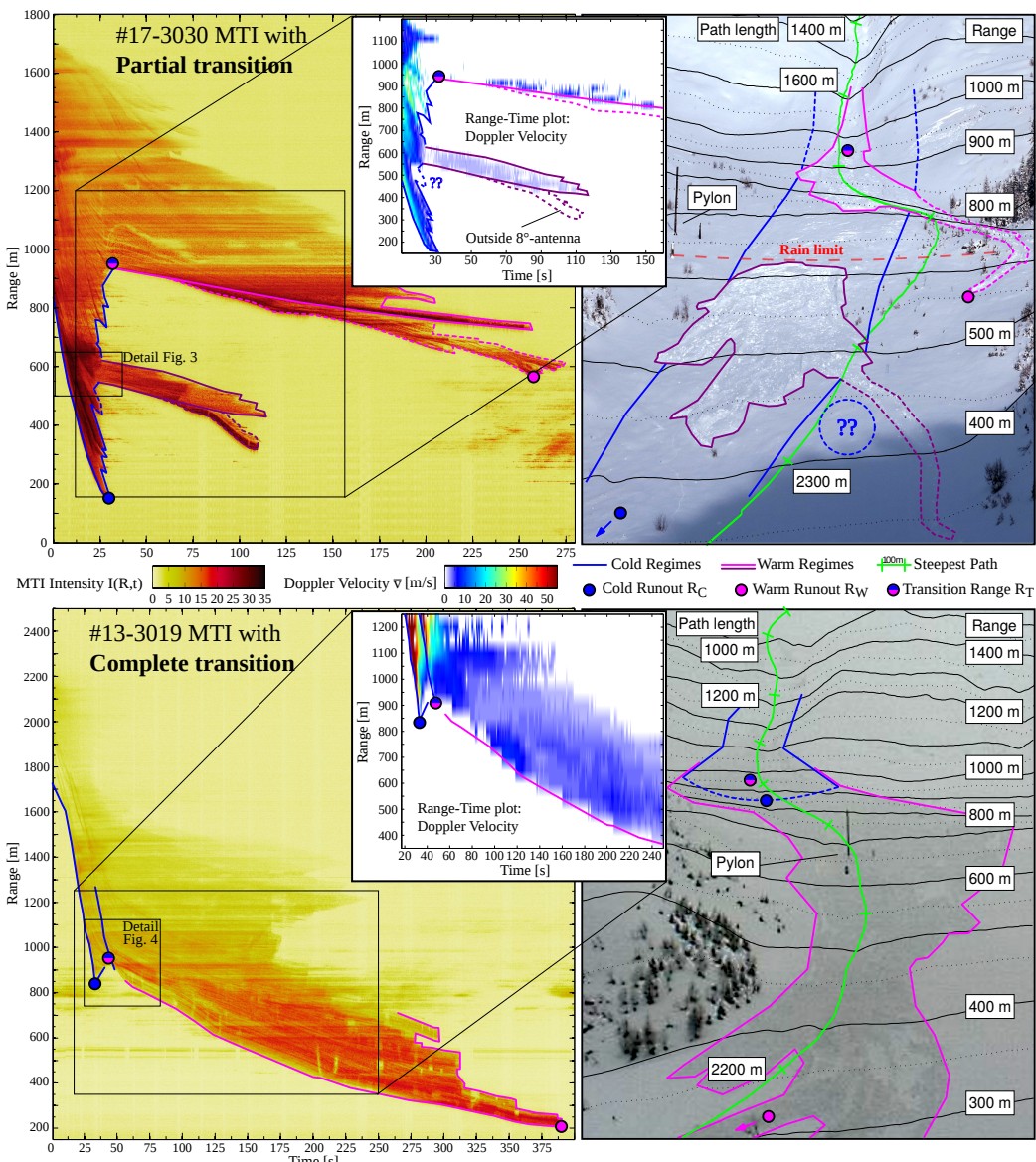

**Figure 2.** Avalanche examples for a partial (top) and a complete (bottom) cold-to-warm transition. The avalanches are visualized by means of GEODAR data (left), mean Doppler velocities $\overline{v}_k(t)$ (middle) and geo-referenced pictures of the deposits (right). Flow features extracted from GEODAR are highlighted in the other panels. The warm regimes are identified by typical coarse-grained and rough deposits (purple and magenta), while the fine-grained and smooth cold deposits can only be sketched (blue). The path along the steepest descent is drawn in green. The cold and warm runout distances and the transition point are indicated with coloured dots.

Since the lower weather station (VDS3) first became operational in the winter season 2012/13, we selected large avalanche events from then until the season 2016/17. From totally measured 130 avalanche events, 18 avalanches fulfill these criteria and were selected. Two of them are compared in detail in Figure 2. The selected avalanches cover the full variability between partial (Sec. 3.1) and complete flow regime transitions (Sec. 3.2). Noteworthy is that avalanches with a complete transition are relatively rare in our data set. There was a three-day period at the beginning of February 2013 when three out of the four of these avalanches occurred. Avalanches with a partial transition could occur all winter from December to March. The avalanche and snow cover data used in this publication are summarized in Table 1.

A release location $[X_0, Y_0, Z_0]$ was assigned to each avalanche event by the use of additional data from the VdlS test site such as photographs and data from flow profiling radars (Köhler et al., 2018). We map the radar range $R$ onto the line of steepest descent from the release location (*i.e.* green line in Fig. 2). Such a procedure can be thought of as a transfer function between radar range $R$, real world coordinates $[X, Y, Z]$ and the path length $P$ (Köhler et al., 2016). The path length $P$ is the projected ground parallel distance from the release point $P_0 = 0\,\mathrm{m}$. Whereas the radar range $R$ is the line-of-sight distance, and is generally smaller than $P$. Since we often do not know precisely the release coordinates, the highest point of the most likely release area was used, giving an uncertainty of $50\,\mathrm{m}$ to $100\,\mathrm{m}$ in path length $P$.

From the MTI plots of the GEODAR data (left in Fig. 2), we extracted manually the following ranges and calculated the corresponding path lengths:

- $P_c$: Path length of front containing cold snow, primarily identified by a stopping with the starving mechanism.

- $P_w$: Path length of front containing warm snow, primarily identified by a backward propagating shock or abrupt stopping.

- $P_t$: Path length until the point of transition between a cold front and a warm front. For avalanches with a complete transition $P_t$ was relatively precise. For partial transitions $P_t$ could be identified only as soon as the warm front separated from the rest of the flow (Fig. 2) and this gave rise to an uncertainty of $\pm 50\,\mathrm{m}$ in path length.

The coloured dots in Figure 2 show the features in the MTI images to which the three points $R_C$, $R_W$ and $R_T$ belong for two example avalanches. The transfer function between radar range $R$ and path length $P$ is roughly given by the labels in the photographs in Figure 2.

## 3  Results

This section starts with a qualitative characterisation of both cold-to-warm flow regime transition types by means of GEODAR and pulse-Doppler data. Then we relate the degree of transition of all 18 avalanches with the snow cover data. Here, we do not differentiate in detail the flow regimes classified by Köhler et al. (2018), but simply consider cold and warm flow regimes only. We call cold regimes those flow regimes which contain cold snow $(< -1\,^\circ\mathrm{C})$, *i.e.* the cold dense regime and intermittent regime. And we call warm regimes those flow regimes which occur for warm snow temperatures $(> -1\,^\circ\mathrm{C})$, *i.e.* the warm shear regime and warm plug regime. Warm and cold regimes differ clearly in their MTI stopping signatures. We refer to Köhler et al.

(2018) for a detailed description of stopping signatures in the GEODAR signal and the differentiation between cold and warm *flow regimes*.

Note, we can not validate the temperature threshold for snow granulation of $-1\,^\circ\mathrm{C}$ with our data. We focus on snow temperature as driving factor, other influences like liquid water content or salt content in maritime snow are neglected. Furthermore, as we use snow cover simulations to examine the temperature of the flowing avalanche, we explicitly assume that the temperatures of the flow and the snow cover is the same. This is clearly an assumption, which depends for example on the entrainment rate, but it is the best we can do.

Figure 2 gives an overview of how cold-to-warm transitions manifest themselves in an MTI image, in the mean velocity from Doppler radar and in a picture of the deposit structures. In the pictures, it is feasible to clearly define the deposits of the warm flow regimes (purple and magenta), while the lateral extent of the cold regimes (blue) can only be sketched. The outlines around regions of the flow regimes can also be extracted from the GEODAR and Doppler data (annotated with the same colors). Due to a smaller opening angle of the Doppler radar antenna, features on the far-right side of the track are not captured (dashed).

When the most distal deposits are cold, a partial transition happened higher up in the avalanche path and deposits of warm snow can be identified (#17-3030, top panels). In contrast, a complete transition happens when an initially cold avalanche starves and transforms into a warm avalanche (#13-3019, bottom panels). While cold regimes are rather quick, warm flow regimes separate in range and time as they are much slower (Fig. 2). The timing when the avalanche reaches the farthest runout distance is therefore different. The avalanche with complete transition (#13-3019) reaches the farthest runout around $350\,\mathrm{s}$ later than the avalanche with partial transition (#17-3030).

## 3.1 Example of a partial transition

The upper panel of Figure 2 shows avalanche #17-3030 as an example of a partial transition. This avalanche originated from the right hand side of the release area and followed the right couloir. The snow consisted of mainly freshly fallen cold snow and was for most of the avalanche track colder than $-1\,^\circ\mathrm{C}$ (upper panels of Fig. 1). The $-1\,^\circ\mathrm{C}$ line was estimated at $H_s = (1416 \pm 77)\,\mathrm{m\,a.s.l.}$ ($\approx 200\,\mathrm{m}$ range) and thus close to the valley floor. Avalanche #17-3030 was a typical powder snow avalanche for the Vallée de la Sionne path, with an intermittent regime at the front and followed by a slow moving dense tail (Sovilla et al., 2015). The geo-referenced picture on the top-right of Figure 2 was taken after $1.5\,\mathrm{days}$ with intense snow fall. Still, the rough deposition patterns of the warm flow regimes can be easily identified, whereas the fine-grained deposits from the cold flow regimes were hidden under the new snow cover.

The GEODAR data are complemented by velocity data captured by the Doppler radar (top-middle in Fig. 2), which shows the mean velocity $\overline{V}(R,t)$ in a range-time plot, *i.e.* the expected value of the velocity distribution for every time $t$ and range $R$ (Eq. 2). Unfortunately, the start of the Doppler radar was delayed by $10\,\mathrm{s}$, thus most of the front is missing, but the regions inside the avalanche where fast and slower flow regimes prevail can be clearly identified. Several fast surges are visible, and were characterized by a velocity of up to $30\,\mathrm{m\,s^{-1}}$. These surges belonged to the cold regimes which can be identified on the basis of their starving stopping signatures. The farthest point reached by the avalanche was the runout of the cold front at

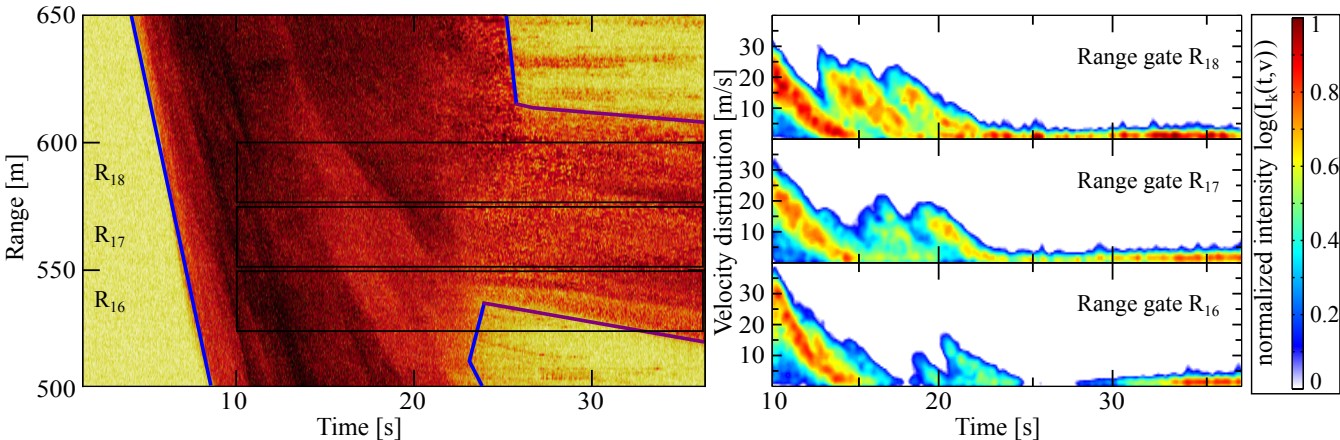

**Figure 3.** Detail of a partial transition from avalanche #17-3030 from the top panels of Figure 2. Left: Zoomed MTI plot with location of Doppler range gates. Right panels: Doppler velocity distribution in the ranges gates $R_{16}$ (525–550 m), $R_{17}$ (550–575 m) and $R_{18}$ (575–600 m).

$R_C = 150\,\mathrm{m}$ range (blue dot, Fig. 2, top), which corresponds to $P_c = 2600\,\mathrm{m}$ path length. This avalanche had a cold-dominated runout.

Two slowly flowing tails followed after the front had passed and were characterized by a homogeneous velocity of $2\,\mathrm{m\,s^{-1}}$ to $5\,\mathrm{m\,s^{-1}}$. Both tails show the characteristic abrupt stopping signatures of warm snow. The transition into the magenta tail

becomes visible in the MTI plot at the end of the steep couloir at a range $R_T = 950\,\mathrm{m}$ (blue/magenta dot). Interestingly, the avalanche's flowing length started to increase already at a range of $1300\,\mathrm{m}$, which suggests that a transition towards the warm and slower regime may have started higher up. However, the warm tail continued to flow for another $250\,\mathrm{s}$ until it finally stopped at $R_W = 550\,\mathrm{m}$ range, corresponding to $P_w = 2140\,\mathrm{m}$ path length. A warm tail like this one is characteristic for most of the powder snow avalanches observed in VdlS. Sometimes, an avalanche can have two of them flowing in both couloirs at

the same time. In this case, the warm runout is defined by the tail which went farthest.

The tail at $400\,\mathrm{m}$ to $600\,\mathrm{m}$ range (outlined in dark purple, Fig. 2) is an unusual feature which we only observe in this data set. However, it enables an excellent opportunity to detail the formation of such a warm tail. It originated from entrainment of warm snow in the 20 degree slope of the runout area. Interestingly, the upper boundary of the entrainment corresponds to a rain limit at $1600\,\mathrm{m\,a.s.l.}$ a few days before the avalanche. The liquid water ingress may have caused a weakening of the snow

cover.

Figure 3 gives a detail of the transition leading towards this warm tail. In the right panels, the velocity distributions of the corresponding range gates $R_{16}$, $R_{17}$ and $R_{18}$ from the Doppler radar are shown. Three surges are visible in these range gates with high velocities at their fronts that decline towards their tails. For the first two fronts, the velocity distribution ranges from $10\,\mathrm{m\,s^{-1}}$ to $30\,\mathrm{m\,s^{-1}}$. The lower signal intensity at smaller velocities indicates that most of the snow moves fast. By

comparison, the approach velocity of the front $v_a$ extracted from the GEODAR data is around $25\,\mathrm{m\,s^{-1}}$. The Doppler data

show that the velocity during the transition changed rather rapidly from fast to slow inside one range gate. Along the three range gates, the first front continues with similar velocity distribution, but the second and third surge diminish. The third front in $R_{18}$ already contains low velocities at its beginning, possibly corresponding to the formation of the warm tail. The terminal velocity (later than $30\,\mathrm{s}$) of the warm tail is characterised by a narrow velocity distribution as expected for a plug-flow regime in all three range gates.

## 3.2 Example of a complete transition

The lower panels of Figure 2 show the GEODAR data, Doppler data and a picture of avalanche #13-3019 as an example of a complete cold-to-warm transition. The avalanche descended from the left hand side and followed the left couloir. The snow cover was wetted by rain up to around $2000\,\mathrm{m\,a.s.l.}$. The temperature pattern was highly dependent on the aspect (bottom left of Fig. 1), but the altitude $H_s = (1989 \pm 74)\,\mathrm{m\,a.s.l.}$ ($\approx 1400\,\mathrm{m}$ range) summarizes the simulated snow cover reasonably well. Avalanche #13-3019 would normally be classified as a warm-wet event, since the deposit showed the typical rough and coarse-grained surface and levées could be identified. But the GEODAR data reveal that a complete flow regime transition occurred at $R_T = 950\,\mathrm{m}$ (magenta/blue dot, bottom left in Fig. 2).

Above the transition at $R_T$, two major surges can be identified with high velocities. The approach velocity $v_a$ measured with GEODAR was $30\,\mathrm{m\,s^{-1}}$ to $35\,\mathrm{m\,s^{-1}}$, while the Doppler data showed material velocities of $50\,\mathrm{m\,s^{-1}}$ to $60\,\mathrm{m\,s^{-1}}$ (Gauer et al., 2007). Such a velocity difference is usually found in the intermittent regime of the frontal region in powder snow avalanches (Sovilla et al., 2018), and corroborates on the turbulent character of both surges (Köhler et al., 2016). The first surge continued for another $100\,\mathrm{m}$ after the transition point $R_T$, and finally starved at $R_C = 840\,\mathrm{m}$ range (blue dot, Fig. 2, bottom), corresponding to $P_c = 1630\,\mathrm{m}$ path length. Note, all avalanches with complete transition in the data set show for the cold front the starving stopping signature. The starving front is a primarily indicator for cold regimes, so that we clearly exclude any other flow regime transition such as warm shear to warm plug transition (Köhler et al., 2018). Therefore the path length of the cold regimes is always farther than the transition point.

Below the transition, the avalanche quickly decelerated and revealed the MTI signature of a warm plug regime – the parallel streaks are interpreted as the signature of large granules riding on a fairly stable surface of the flow due to a homogeneous velocity field (Köhler et al., 2018). The mean velocity decreased after the transition to around $3\,\mathrm{m\,s^{-1}}$ to $5\,\mathrm{m\,s^{-1}}$ and was very homogeneous in the full body of the avalanche (bottom-middle in Fig. 2). The warm flow regime continued to flow for another $300\,\mathrm{s}$ before reaching the farthest runout at $R_W = 200\,\mathrm{m}$ range (magenta dot) and $P_w = 2370\,\mathrm{m}$ path length. This avalanche thus had a warm-dominated runout.

Figure 4 shows a zoom of the transition region as an MTI image (left) and distributions of the Doppler velocity in three range gates (right). In $R_{18}$, the front of the first surge showed low intensity for small velocities, but a broad spectrum of velocities between $20\,\mathrm{m\,s^{-1}}$ to $70\,\mathrm{m\,s^{-1}}$. The second surge was in general slower, and showed large intensities in a narrow and slow velocity band. The MTI image indicates that streak signatures (black) crossed the second surge and suggests that the low velocities belonged originally to the first front. The duration of the high velocity region in each surge was rather short with $5\,\mathrm{s}$, compared to fully developed powder snow avalanches where this region can last up to $40\,\mathrm{s}$ (Steinkogler et al., 2014).

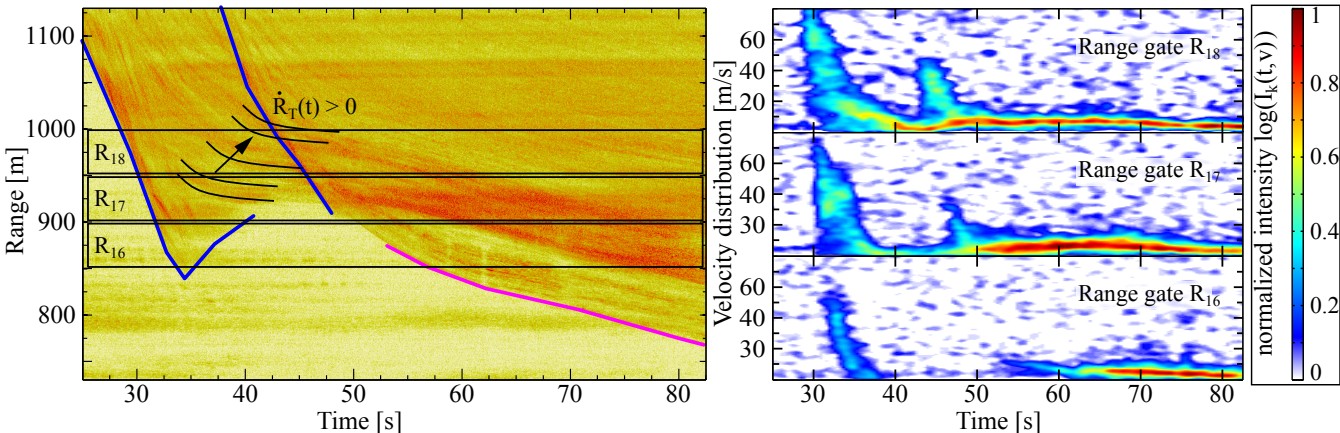

**Figure 4.** Detail of complete transition from bottom panels of Figure 2. Left: MTI with location of Doppler range gates. The location of transition $R_T$ is not fixed, but moves upward with time $\dot{R}_T(t) > 0$. Right panels: Doppler velocity distribution in range gates $R_{16}$ (850–900 m), $R_{17}$ (900–950 m) and $R_{18}$ (950–1000 m).

However, the velocity distribution after the transition was narrow with the centre at low and constant velocity indicating a plug flow. Interestingly, the velocity distribution in the plug regime showed very little intensity for velocities between zero and 2–3 m s$^{-1}$, which indicated a very coherent movement of the avalanche (Fig. 4, Doppler data $R_{17}$ and $R_{18}$ at t >50 s).

The flow regime transition happened rather quickly in this avalanche as well as in the other avalanches with complete transition in our data set. The transition occurs within around 100 m travelled distance and over a period of less than 15 s. Furthermore, the location of the transition seemed to have traveled uphill ($\dot{R}_T(t) > 0$) as the black lines in the left of Fig. 4 indicate. Note, no material is traveling upwards at the transition point, but the shock front of the deceleration is moving. This may be caused by a piling up of incoming fast material on top of the already decelerated material. Or, an alternative explanation could be that material flowing into the range gate later is already slower and therefore stops more easily at higher locations. However, a complicated model-based dynamic interpretation of the MTI plot and the Doppler data would be needed to decide between both possible interpretations.

As in avalanche #17-3030, the flowing length started to increase at a range of 1500 m (bottom-left Fig. 2), indicating a separation of fast and slow material in direction of the flow. Faster and possibly cold material may have been concentrated towards the front, while slower and maybe warm material segregated towards the tail.

## 3.3 Snow cover influence on transition type

To differentiate between avalanches with partial and complete transitions, we quantify the degree of transition by defining the transition index

$$F_t = \frac{P_W - P_C}{\max(P_C, P_W)} \tag{5}$$

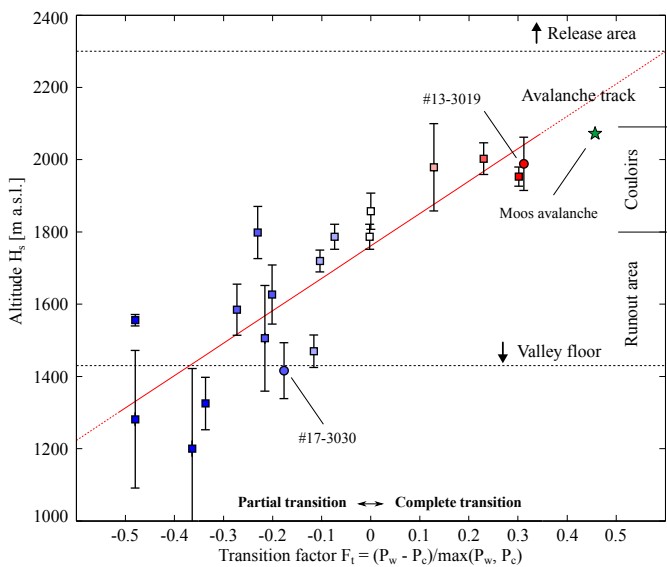

**Figure 5.** Transition index $F_t$ as a function of $H_s$ with a linear regression in red. The transition index $F_t$ has an uncertainty of $\pm 0.05$ to $0.1$. Green star belongs to the Moos avalanche mentioned in the introduction. Horizontal dashed lines and annotation on the right side characterizes roughly the VdlS terrain.

as the difference between the path length from cold ($P_c$) and warm ($P_w$) flow regimes divided by the total path length reached by the avalanche. For avalanches with a partial transition (*e.g.* Sec. 3.1), the transition index is negative and the runout is dominated by cold regimes. For events with $F_t \approx 0$ the cold regime and the warm regimes reach the same runout. For a positive transition index, the runout is dominated by warm regimes, corresponding to avalanches with a complete transition

(*e.g.* Sec. 3.2). A value of $\pm 0.5$ means that the dominant regime reaches twice as far as the other regime. The limits of $F_t$ to both sides, *i.e.* $F_t = -1$ and $F_t = 1$, correspond to avalanche types made of purely cold regimes and purely warm regimes, respectively. The avalanches from the examples in Figure 2 have transition indices of $F_t = -0.18$ (#17-3030) and $F_t = 0.31$ (#13-3019). Note, we do not give an uncertainty of the transition index $F_t$ explicitly in Table 1. However, an uncertainty of $50\,\mathrm{m}$ to $100\,\mathrm{m}$ in the path lengths $P_c$ and $P_w$ propagate into the transition index $F_t$ as an uncertainty of $\pm 0.05$ to $0.1$.

The transition index $F_t$ together with the altitude $H_s$ for all avalanches are shown in Figure 5. The 18 analysed avalanches cover $F_t$ in the range between $-0.5$ and $0.4$, and the set of values is well distributed over this range. A linear regression gives $H_s(F_t) = (895 \pm 149) \cdot F_t + (1760 \pm 39)$ with a correlation coefficient of $r = 0.85$. For pure warm avalanches ($F_t = 1$), the regression gives $H_s$ at $2660\,\mathrm{m\,a.s.l.}$, which corresponds to the altitude of the release area. For pure cold avalanches ($F_t = -1$), the regression would give $H_s$ at $860\,\mathrm{m\,a.s.l.}$ which is far below the runout area of Vallée de la Sionne at $1400\,\mathrm{m\,a.s.l.}$ However,

we do not think that the extrapolation towards purely cold avalanches ($F_t = -1$) has any validity in this setting.

Figure 6 compares the altitude $H_s$ against the altitude $H_t$, that is where the snow cover changes from $-1\,^{\circ}\mathrm{C}$ against where the transition occurs. We find the altitudes of the transitions $H_t$ scatter on both sides of the 1:1-line (blue dashed); in other

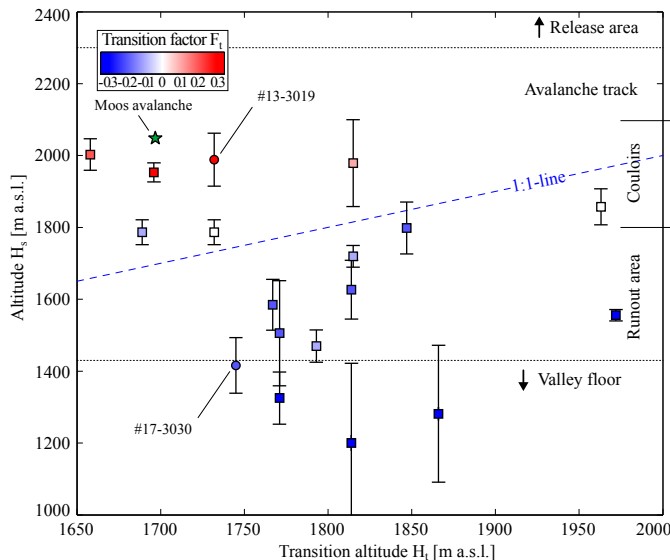

**Figure 6.** Altitude of transition, $H_t$, against the altitude of the $-1\,°C$ line, $H_s$. The 1:1–line (dashed blue) divides the avalanches into cases where the transition occurs above or below the $H_s$-line. Horizontal dashed lines and annotation on the right side characterize roughly the VdlS terrain. The colour indicate the transition index from Figure 5.

words, the transition can happen above or below the $H_s$-line. Furthermore, $H_t$ can be up to $500\,m$ in elevation away from $H_s$. The majority of the avalanches perform the transition above the $H_s$-line, *i.e.* avalanches with a partial transition (blue symbols). For these events we find that $H_s$ lies below $1800\,m$ a.s.l. and thus in the runout area. And for a few of them, the $H_s$-line is even below the valley floor (below $1450\,m$ a.s.l.), which in turn means that it can practically not be reached and that entrainment of
surface snow can not cause a partial transition.

The remaining avalanches perform the transition below the $H_s$, *i.e.* these events express either a complete transition and the warm regimes are dominant in the runout (red symbols), or cold and warm regimes reach similar runouts (white symbols). For these events we find that $H_s$ is consequently higher than $1800\,m$ a.s.l. which corresponds approximately to the altitude of the middle of the avalanche path so that the entrainment of surface snow increases the avalanche temperature.

**4  Discussion**

**4.1  Discussion of results**

We find a continuous degree of transition between partial and complete flow regime transitions (Fig. 5). This continuous degree can be related to the altitude $H_s$, the altitude where the average modelled temperature of the surficial snow layer changes from below to above $-1\,°C$. This means that, the flow regime type in the runout area — but not the runout distance itself — can be
estimated when $H_s$ is known.

This (semi-)quantitative attempt to capture an aspect of the flow regime transitions with a minimum number of observable quantities need further investigations to find out if the proposed linear fit or other relations are valid. Even if the linear fit appears to work at least for a certain range of $F_t$ values, one still needs to find answers to the asymptotic behaviors when $F_t$ tends towards $-1$ or $1$. Similar, we can not test the path dependency of our results since all our data are from the VdlS avalanche path. However, we think that at least the following three limitations are important to bear in mind for the discussion of the results.

- The transition index will probably be most useful for avalanches with drop heights of more than $500\,\mathrm{m}$. For smaller avalanches, $H_s$ tends to be either above the release area or below the run-out area.

- While $H_s$ can be determined wherever and whenever there is enough meteorological data for running snow-cover simulations, finding $H_t$ for a given event requires either detailed investigation of the avalanche deposits or measurements with a GEODAR or Doppler radar.

- For use as a predictive tool, e.g. for road closures or evacuations, a plot like Figure 5, containing many events, would be necessary. Probably, such copious and detailed data is available only for a handful of avalanche paths worldwide yet.

Avalanches with a cold-dominated runout occur in Vallée de la Sionne when $H_s$ is up to $300\,\mathrm{m}$ in elevation above the valley floor. The nomenclature of UNESCO (1981) would classify such an avalanche as "C1G7", with the code 7 meaning the deposit consists of a mix of cold-dry and warm-wet snow. We find that the point $H_t$ where the transition becomes visible lies exclusively above $H_s$ for cold-dominated avalanches. Thus the transition cannot be caused by snow erosion from the surface, but entrainment of deeper and therefore warmer layer of the snow cover must be accounted for. Since the surface (*i.e.* new snow) is cold, a powder snow avalanche maintains its dynamics from surface entrainment, but later flowing parts like the denser core may eventually dig deeper into the snow cover, erode the warmer snow layers and develop a warm tail even above $H_s$.

We observe that nearly every large powder snow avalanches in Vallée de la Sionne undergoes a partial transition. This suggests that large purely cold-dry powder snow avalanches are very rare. In all GEODAR data acquired over the last 7 years (140 in total with 20 powder snow avalanches), only one large powder snow avalanche (#15-0017, Köhler et al. (2016)) without a clear partial transition can be found. This avalanche was released shortly after avalanche #15-0016 ($F_t = -0.48$) which had entrained and removed most of the snow in the track. Purely cold-dry avalanches do exist, but perhaps, only as long as they stay small and thus entrain only layers of cold snow close to the surface.

Warm-dominated avalanches are usually classified as wet avalanches, since such a description is mostly based on the deposit texture. Our data show that initially cold-dry avalanches can produce completely warm-wet deposits ($F_t > 0$). A special nomenclature for those avalanches does not exist or is not used consistently, even though the UNESCO avalanche classification scheme allows for different wetness classes in the release and runout areas. An avalanche with a complete transition could be denoted as "C1G2" (UNESCO, 1981). The results in Figure 5 indicate that such avalanches occurred in VdlS when $H_s$ is more than $500\,\mathrm{m}$ in altitude above the valley floor. We find that $H_t$, the point where the transition is initiated, is consequently $200\,\mathrm{m}$

to $300\,\mathrm{m}$ below $H_s$ (Fig. 6). This indicates that entrainment of warm snow from the surface is most likely the cause for the transition, but also that a previously developed cold flow regime may be able to overflow a surface of warm snow for about this distance. As soon as the transition towards warm regimes begins, it happens instantaneously and not gradually, *i.e.* in only $100\,\mathrm{m}$ and $15\,\mathrm{s}$ (Fig. 4).

Interestingly, the actual altitude of the transition $H_t$ differs for events with partial and complete transitions (Fig. 6). All partial transitions in cold–dominated avalanches occurred in the elevation band between $1750$ and $1850\,\mathrm{m\,a.s.l.}$, which corresponds to the altitude at the end of the steep couloir. Complete transitions could occur even at lower elevations down to around $1650\,\mathrm{m\,a.s.l.}$, which correspond to the gently inclined runout area and even the altitude of the pylon. We think that the above mentioned change in the terrain does not necessarily cause the transition, but gentle terrain may favour the warm and presumably slower flowing snow to separate from fast cold regimes in flow direction. Such a separation can be observed at higher elevations where the flowing length starts to increase and the avalanche extends in range in the MTI plots (Fig. 2). This lengthening occurs most often above $H_t$ and may indicate an earlier start of the transition and a separation of slower and faster flowing regions.

Both transition types are relevant for the dynamics at the avalanche front and especially during deposition in the runout area. For partial transitions, the relevance is indirect as the runout is still cold-dominated, but the slow warm tail keeps mass away from the front and reduces the size of the cold flow regimes. For complete transitions the relevance is obvious, as the runout is warm-dominated even though a cold avalanche released. The time-scale when a warm-dominated avalanche reaches the runout is delayed by several hundreds of seconds due to slower velocities of the warm flow regimes (Figure 2). More importantly, the pressure exerted on structures in the runout depends strongly on the flow regime, and in general is a function of velocity, density and flow height together with a geometry factor (Sovilla et al., 2016). Cold–dominated flow regimes have a dominant velocity squared contribution and the hydrostatic term vanishes due to small densities. In contrast for warm flow regimes, the dynamic term can be neglected due to smaller velocities, but the large density increases the importance of the hydrostatic pressure contribution. Sovilla et al. (2016) presented an example which deviates from the cold or warm pressure scheme and both — dynamic and hydrostatic — contributions are found to be important. We can imagine that avalanches with a complete transition may generate similar high pressures during the transition process as result of remnant high velocities together with an increase in density. Such an argument seems to be different for avalanches with a partial transition. As mentioned above, the warm tail results most likely from deep entrainment by the dense core where the velocities are slower than at the front, so that the dynamic pressure contribution probably stays small.

Another important topic is the extent to which frictional heating due to dissipation processes during the avalanche descent may play a role in flow regime transitions (Vera Valero et al., 2015). Frictional heating compared to a temperature increase due to entrainment was recently investigated experimentally on two medium sized purely cold avalanches by Steinkogler et al. (2015b). They concluded that frictional heating depends mainly on the effective height drop, but the contribution due to entrainment was found to be more variable and dependent on the erosion depth and snow temperature. Here, we cannot differentiate between both heating mechanisms on the basis of our data set. In fact, we include the frictional heating of the

flowing snow as it affects $P_w$ and $P_c$ indirectly. However, the relation in Figure 5 indicates that indeed snow erosion and the temperature of the eroded snow have an important effect on the flow dynamics.

## 4.2 Limitations of methodology

Two limitations in regard to temperature exist in our methods. Throughout the whole study, we have assumed that the flowing snow temperature is similar to the snow cover temperature. This is a vague and untested assumption, and the effect depends possibly on the entrainment rate and the temperature difference between the flowing snow and the snow cover. This assumption does not affect the correlation between $H_s$ and $F_t$ observed in the data. However, it would be an important factor for a generalization of the presented empirical approach. Furthermore, the history of avalanche activity in the avalanche path can significantly alter the snow cover by entrainment and deposition (Steinkogler et al., 2014). The `SNOWPACK` model can account for this with reinitialization of the snow cover. But this can be only done for artificial avalanches where precise mass-balance measurements are available. Our approach disregards this fact. However, we are interested in the surface layers consisting of the recent new snow precipitation. The simulation of these new top layers is more dependent on the meteorological data than on the older snow layer underneath.

Also questionable appears the estimation of $H_s$ by linear interpolation between two weather stations. We imply that the snow temperature changes only due to an altitude gradient, and this altitude gradient is found to be in the range of $100\,\mathrm{m}\,^\circ\mathrm{C}^{-1}$ to $400\,\mathrm{m}\,^\circ\mathrm{C}^{-1}$. The estimate of $H_s$ could be improved with detailed analysis performed with distributed snow cover models like `Alpine3D`. Such analysis has been done by Steinkogler et al. (2014), but their result show small deviation from linear along the Vallée de la Sionne avalanche path. However, we wanted to use a simple parameterization for $H_s$. "Simple" means that $H_s$ can be estimated from different data sources, *e.g.* field observations or regional snow reports, since for many avalanche paths and past events much less information about the snow cover characteristics is generally available.

The presented method is based on the temperature threshold of $-1\,^\circ\mathrm{C}$ with respect to the uppermost $0.5\,\mathrm{m}$ of the snow cover. To validate this $-1\,^\circ\mathrm{C}$ temperature threshold is not the scope of this study, but we have tested our method also for $-2\,^\circ\mathrm{C}$ and $-0.5\,^\circ\mathrm{C}$. The effect is a shift in the $H_s$ altitude, *e. g.* in Figures 5 and 6. Our results indicate that $-1\,^\circ\mathrm{C}$ is a reasonable value. For $-2\,^\circ\mathrm{C}$ the partial (blue) and complete transitions (red) are not split anymore by the 1:1-line in Figure 6. And for $-0.5\,^\circ\mathrm{C}$, the regression in Figure 5 predicts for pure warm avalanches ($F_t = 1$) only a $H_s$ altitude of $2150\,\mathrm{m}$ a.s.l., which is clearly below the release areas so that regions with cold flow regimes are expected.

Another difficulty is how to generalize our results to other avalanche tracks since we have only investigated a single slope. We expect a path dependence of the correlation between snow cover and the transition index $F_t$. Vallée de la Sionne is known to be a relatively gentle avalanche path so that avalanches normally stop naturally in the runout area. But for steeper paths, *i.e.* $40°$ from top to bottom, we expect that more often both flow regimes may reach the valley floor. Our analysis should be extended to take into account other variables, such as volume or mass estimates and path geometry. To directly extend our method to other avalanche paths, regional snow and avalanche reports as well as path length estimation from world-wide available digital terrain models, may already be sufficiently accurate. As an example, the Moos avalanche from the introduction fits into the

relation found for VdlS (star in Fig. 5 and 6), but noteworthy to say, the geometry of this avalanche path in terms of altitudes, slope and path length, is very similar to the VdlS.

## 5 Conclusions

GEODAR measurements have shown that flow regime transitions are common in large snow avalanches. One of these transitions occur between cold and warm snow when agglomeration of snow grains causes larger granules to form. In first order, this happens as soon as the flowing snow temperature changes from below to above $-1\,°C$. Such a flow regime transition is very important for the dynamics of the avalanching snow, as the flow regime influences the flow mobility and the pressure exerted on structures in the path. However, we want to stress that the runout distance itself does not depend on the flow regime as cold and warm avalanches can reach unexpectedly long runouts.

We find two types of cold-to-warm flow regime transitions depending on whether parts or the complete avalanche changes the flow regime. A partial flow regime transition can occur at the tail and depends on the entrainment of deeply buried warm snow layers by the avalanche's dense core. In contrast, a complete flow regime transition can occur at the front due to the entrainment of warm snow at the surface. We find a continuous degree of transition between both types and a relation between this and the snow cover temperature along the avalanche track. More specifically, the transition index $F_t$ is linearly related to the altitude $H_s$ where the average snow cover temperature in the uppermost $0.5\,m$ changes between warm and cold at a threshold of $-1\,°C$.

At Vallée de la Sionne, almost all large powder snow avalanches exhibit a transition. Given the choosen assumption of threshold temperature of $-1\,°C$ measured in the uppermost $0.5\,m$, we find when $H_s$ is found no higher than $300\,m$ above the valley floor, a partial transition ($F_t < 0$) is observed and results in a warm tail. For complete transitions ($F_t > 0$), the altitude $H_s$ is located more than $500\,m$ above the valley floor and results in only warm flow regimes in the runout area.

This work can be regarded as a first step in developing a method for predicting the dominant flow regime in the runout area — but not the runout length — based on knowledge of the snow cover temperature along the path. It is worth mentioning that meteorological and snow cover data from the release area are not representative for the avalanche dynamics in the runout area. Therefore, any hazard and risk evaluation should be made with additional information. Knowing the flow regime in the runout area may improve risk assessment, for example, the effectiveness of a dam may be evaluated in real-time. Nevertheless, the presented approach is strongly dependent on the track geometry and this requires care in adapting our results to other avalanche paths.

Compared to the complexity of temperature influence on avalanche dynamics, our presented method is rather simple. Effects such as frictional heating, temperature difference between entrained and flowing snow, entrainment depth and mixing and separation of snow at differently temperatures are important factors, and to identify their significance on the flow dynamics is a challenging task. We are convinced that future measurement procedures with laser-scans for mass balance, infrared radiation thermography in combination with temperature measurement during the passage of an avalanche, and manual or simulated snow profiles will be very useful to further understand the interplay between these factors. Finally, investigating flow regime

transitions in greater detail may become important in respect to climate change. Less snow cover at lower altitudes, strong temperature gradients and quickly varying weather systems may lead to a snow cover situation favouring transitions in avalanches. Warm flow regimes may reach runout areas more frequently and thus require that hazard mitigation procedures be adapted accordingly.

5 *Data availability.* The data used in this publication is available upon request to the corresponding author. Most of the GEODAR data can be sourced from the GEODAR data repository (McElwaine et al., 2017).

*Competing interests.* The authors declare that they have no conflict of interest.

*Acknowledgements.* The research was funded by the Swiss National Science Foundation (SNSF) project "High Resolution Radar Imaging of Snow Avalanches," grant 200021_143435. Special thanks are due to our colleagues at the electronics and workshop of SLF for their
10 invaluable support. We are grateful to Dieter Issler, Alexander Densmore and one anonymous reviewer for their valuable comments which increase the clarity and quality of this paper.

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
