# Peer review of "Cold-to-warm flow regime transition in snow avalanches"

_The Cryosphere, 2018_

## Referee Comment (RC1) · Anonymous Referee #1 · 29 Jun 2018

The paper by Köhler et al presents detailed measurements obtained from radar techniques (the GEODAR, already presented in a number of previous papers, and the pulse-Doppler system), conducted at Vallée de la Sionne avalanche test site, Switzerland. The authors pay attention to the cold-to-warm transition during flow propagation, by considering in parallel the temperature calculated from the snow cover model SNOWPACK. The introduction provides some general ideas on the problem of wet-snow avalanche and more particularly on the cold-to-warm transition problem with some relevant recently published papers. Section 2 presents a brief description of the two radar techniques used and the snow cover reconstruction with SNOWPACK, as well as the data sets. Section 3 describes the results by distinguishing between complete and partial warm-to-cold transitions. It is first based on one example for

each transition and two key graphs (Figs. 5 and 6) including all the avalanche events considered are then presented and shortly discussed. Section 4 is a more extended discussion on the results. The limitations of the study are discussed too. Finally, section 5 concludes the manuscript by synthesizing the main results and discussing future works to be done.

General comment:

The present paper relies on radar techniques that are advanced in situ measurements methods to "look inside the avalanches", following a number of recent studies (about GEODAR in particular). By coupling those radar measurements with snow cover reconstruction (the temperature in particular) and making use of the rough assumption that the temperature of the flowing snow is equal to the snow cover temperature, the authors are able to highlight a relation between the degree of cold-to-warm transition (partial versus complete) and the altitude where the snow cover temperature is $-1°C$. Though $-1°C$ has been previously identified as a threshold temperature controlling the transition between nearly no granulation and efficient snow granulation (see the controlled experiments done by Steinkogler et al, 2015), it may appear as an arbitrary threshold.

I enjoyed the reading of the paper. The result shown on Fig. 5 is quite remarkable. The topic addressed in the manuscript is timely. I believe that the manuscript can deserve publication if the authors make an effort to revise some points. The success of Fig. 6 is somewhat counter-balanced by the result shown on Fig. 6. My main concern on the scientific content is that a sensitivity analysis to the choice of some thresholds (thickness of 0.5 m for the snow cover taken into account, temperature threshold of $-1°C$) is missing. Including such a sensitivity analysis to changing those thresholds is needed in order to reinforce the arguments provided by the authors on the physics of the cold-to-warm transition. How the plots shown on Fig. 5 and 6 would be changed by choosing other values of those thresholds?

I would have a request on the organization of the paper, in addition. The discussion section (section 4) is not well-organized. I invite the authors to make it much more readable. A couple of points that are direct interpretations of key plots shown on Figs. 5 and 6 should be discussed in more detail and moved to section 3.3. The discussion section could be split into two sub-sections: one for a general discussion on the main results and one about the limitations of the methods used.

Please find below a detailed list of major to more minor comments on the manuscript.

List of major/minor points:

Sec. 1 - Introduction

- page 1, line 22: [..., whereas dense flow regimes, especially warm regimes, can be diverted or even stopped.]. This sentence is somewhat reductive. I agree that rapid dry- and cold-snow avalanches are difficult to divert and stop. But some flow regimes of wet-snow avalanches can pose serious problems too. Their interaction with protection structures is sometimes very complex, due to nearly unpredictable flow trajectories around avalanche dams (see some examples in Johannesson et al., 2009; European Handbook, chapter 8). Could you please qualify your statement.

- p. 2, lines 9-10: [when liquid water is still expected to be absent.]. I would remove this statement given the fact that it is now well-established that localized melting can occur at ambient temperatures a few degrees below the freezing point (Dash et al, RMP 2006; Turnbull, PRL 2011).

- p. 2, end of line 10: the existence of that quasi-liquid layer in flowing snow has two consequences. It can increase snow cohesion on the one side (and thus increase the size of aggregates) but it may also lubricate the contacts between snow aggregates on the other side (thus enhance flow mobility). Maybe the second effect could be shortly discussed, in addition.

- p. 2, lines 20-21: [A partial transition affects only the tail of the flowing avalanche and

the final run-out is still cold-dominated.]. This sentence suggests that the transition does not occur at the front but mostly at the tail. Could it be that such a scenario with the cold-to-warm transition occurring at the front does exist?

- p. 3, lines 13-14: why this arbitrary value of 0.5 m? The statement [Despite the crudeness of this measure, we assume that...] needs clarification. Please could you justify? Maybe you could explicitly refer to the paper section which addresses in detail the assumptions made.

- p. 3, lines 20-21: [Finally, the discussion (sec. 4) brings together both result parts and the study is finished by a conclusion (sec. 5).]. This sentence is quite easy: I guess you could propose a more precise sentence, more relevant to the content of your paper in order to announce both 'discussion' and 'conclusion' sections.

Sec. 2 - Methods and data

- p. 4, line 15-16: maybe you could give (at least) one relevant reference already published for each system, the older system and the newer system.

- Fig. 1, caption, line 3 (p. 5): 'shown' (not 'show')

- p. 6, lines 12-23: this part justifies your assumptions made (in particular the 0.5 m). Please refer to this part at p. 3, line 13, in the introduction (see a previous comment).

- p. 8, Table 1: could you please provide an order of magnitude of the error/uncertainty on $P\_c$ and $P\_w$? And thus $F\_t$?

- p. 9, lines 10-13: is there any uncertainty on this threshold of -1°C between warm and cold regimes? Temperature is certainly a very important control parameter but other factors may come into play. Maybe you could discuss this a bit (see another comment below, on Fig. 6).

Sec.3 - Results

- p. 12, lines 3-12: this is a very interesting observation, providing a quantitative proof

of a mechanism known from the field experience gained by some snow avalanche experts. Under a context of climate change / global warming, we may expect more events with rain occurring at high altitude on the snow cover during winter. Your measurements are relevant to this problem. Maybe you could add a short word on this point here.

- p. 12, lines 21-24: [... This discrepancy corroborates the turbulent character of both surges.]. Could you please explain better what you mean here? Do the differences stem from different positions of the devices and/or assumptions made with respect to main flow direction? As such, very turbulent flows, with significant velocities in all (3D) directions can produce different results depending on the technique used. This part needs more clarification.

- p. 13, Eq. (5): could you please give an uncertainty on $F_t$? (back to a previous comment on uncertainties on $P_w$ and $P_c$). And report this uncertainty on Fig. 5.

- Fig. 5: it is nice to see this correlation between $H_s$ and your $F_t$. Would be nice too to study the sensitivity of the plot to changing the threshold of -1°C. Would that plot be improved or deteriorated by choosing a different temperature threshold (below or above -1°C)?

- p. 14 - 15. That you use the linear fit to extrapolate and obtain the value of 860 m a.s.l. for $F_t$=-1 is questionable to me. Because it does concern the arrest conditions of the avalanche, I guess the effect of local topography coupled with the snow (flowing/deposited and entrained) properties is crucial. I would suggest that either you don't extrapolate or your provide more critical discussion on that result.

- p. 15, lines 3-6, and Fig. 6: I may interpret this plot showing $H_s$ versus $H_t$ as a proof that (i) the -1°C threshold may a bit arbitrary and (ii) other factors come into play. Those points need more critical discussion. Maybe some arguments given in the discussion should be already developed here (see another comment thereafter).

Sec.4 - Discussion

- p. 16, lines 10-11: that the flow regime in the run-out zone can be estimated when $H_s$ is known relies on the linear fit proposed for the relation between $F_t$ and $H_s$. You could be more precise here, and add at the same time that this will need further investigation: linear fit or other relation? range of $F_t$ for which the linear fit is valid? asymptotic behaviors when $F_t$ tends towards -1 or +1?. See also a previous comment.

- a general comment: this section is difficult to read because there are too many ideas. I would propose to put some points (in particular: entrainment at the surface versus deeper in the snow cover, effect of the topography, front dynamics) earlier in Sec. 3 and maybe extend the discussion on those points in Sec. 3, because they are direct and important interpretations of the plots shown on Figs. 5 and 6. Also, the remaining points (not transfered to sec. 3) could be a sub-section 4.1 and the discussion on the limitations of the method (starting from line 26, p. 17) could be a sub-section 4.2.

Sec. 5 - conclusions

- p. 18, line 26: the flow regime influences not only the pressure on structures but also the flow mobility (run-out: velocity and volume). Please add those points.

- p. 18, line 32: please remove "robust relation" but (for instance) use "correlation" instead or keep "relation" only. I agree that this result is very nice but this result will need further validation.

- p. 19, lines 3-6: how those values of 300 m and 500 m depend on some (arbitrary) choices you made? A sensitivity analysis (of plots on Fig. 5 and Fig. 6) to changing the threshold values for the temperature (-1°C here) and the snow cover thickness taken into account (0.5m) is missing in your study.

---

## Referee Comment (RC2) · D. Issler (Referee) · 10 Jul 2018

**Content of the paper**

Over a period of some three decades, our capability of numerically simulating the evolution of the snow cover in some detail in 3D has developed to a level where these tools can be used in diverse applications with some confidence (notwithstanding major residual problems). Greatly developed and diversified experimental techniques—all of them installed at the Vallée de la Sionne test site in Switzerland—have given us an unprecedented, detailed view of the processes inside flowing avalanches. One of the most conspicuous of these new instruments is GEODAR, a phase-array-based radar system that eventually will allow high-resolution 3D mapping of entire avalanches through

time. Finally, thanks to IR imaging, the role of the snow temperature in the snow cover and the flowing avalanche has become a major focal point of research in avalanche science, mostly due to work at SLF in Switzerland.

In this paper, the authors combine these three major elements—snow-cover simulation, 18 avalanche measurements with GEODAR and Doppler radar, as well as previously gained insight into the critical role of snow temperatures near melting on the flow regime of avalanches. The main objectives are (i) to verify the effect of snow-cover temperature by comparing a rather large sample of measured avalanches and (ii) to quantify in a simple way how the degree of transition from a dry-snow flow regime to a wet-snow flow regime depends on the snow-cover temperature along the avalanche path.

The measurements with GEODAR and Doppler radar allow distinction between different flow regimes, as recently shown in a different paper by some of the same authors. Moreover, they can distinguish "complete" from "partial" transitions and locate a representative transition point. From the run-out distances of the warm and cold parts of the flow, they construct a transition index $F_t$ and relate it, for each event, to the reconstructed altitude $H_s$ where the mean temperature of the top layers of the snow cover reached $-1°$C. They find a clear positive correlation between $F_t$ and $H_s$. Similarly, warm avalanches (undergoing a full transition) are shown to make the transition to the warm-snow flow regime above the altitude $H_s$, whereas that point is below $H_s$ for all cold-snow avalanches (with partial transition only).

**General comments**

The transition index proposed by the authors is a clever attempt to (semi-)quantitatively capture an aspect of the flow-regime transition process with a minimum number of observable quantities. In order to link it to the thermal regime of the flow, the transition altitude, $H_t$, is invoked and statistically compared to the altitude $H_s$ below which the upper snow cover is warm. The authors seem to be aware of the difficulties and limita-

tions of this approach, but it might be useful to spell them out more explicitly. From my point of view, the following points are particularly important:

- The transition index will probably be most useful for avalanches with drop heights of 500 m or more. For smaller avalanches, $H_s$ tends to be either above the release area or below the run-out area.

- While $H_s$ can be determined wherever and whenever there is enough meteorological data for running snow-cover simulations, finding $H_t$ for a given event requires either detailed investigation of the avalanche deposits or measurements with a GEODAR or advanced Doppler radar.

- For use as a predictive tool, e.g. for road closures or evacuations, a plot like Fig. 6, containing many events, would be necessary. Probably, such copious and detailed data is available only for a handful of avalanche paths worldwide.

That being said, I agree, however, that the transition index and the correlation between $H_s$ and $H_t$ are a meaningful way of demonstrating the relevance of the thermal regime for the flow of avalanches.

The method for determining the uncertainty in the snow temperatures $\bar{T}_{2,3}$ remains unclear to me. The way I read the text, they calculate the standard deviations as $\sigma_T = [\sum_1^N (T_i - \bar{T})^2/(N-1)]^{1/2}$, with the sum extending over the computational layers used by SNOWPACK in the top 0.5 m of the snow cover. If this is indeed what they mean, I cannot see how this should be connected to the uncertainty of $H_s$—that uncertainty is more directly connected to the question whether a linear extrapolation of snow pack temperatures is admissible. As a consequence, I cannot assess whether the authors' approach for determining the consequent uncertainty in $H_s$ is sound or too optimistic. The way they do it according to Fig. 1 assumes that the deviations of $T_2$ and $T_3$ from their means $\bar{T}_2$ and $\bar{T}_3$ are tightly and positively correlated. If this is not the case, the uncertainty in $H_s$ will be much larger. This would, however, have considerable

importance for Figs. 5 and 6 and for the firmness of conclusions that can be drawn from them. These issues have also been commented upon by the other reviewer and need to be addressed carefully by the authors.

The title of the paper is more general than its content in that dynamical aspects are more or less completely left out. However, the GEODAR data offer a unique opportunity to quantify some aspects of the dynamics: Figures 3 and 4 suggest that a major component of the avalanche first moves at a nearly constant speed, then decelerates over a period of 5–10 s, and then continues again at nearly constant speed. From the curvature of the streaks, it should be relatively straightforward to extract the deceleration, and since the location is also known fairly precisely, the retarding accelerations before, during and after the cold-to-warm flow-regime transition can be determined. This is a rather remarkable phenomenon with far-reaching consequences for modeling the flow. I do not understand why the authors hardly mention this, and I strongly encourage them to dedicate a subsection or a few paragraphs to an at least preliminary analysis.

**Minor points**

**P1, L4,5:** The sentence "The intake of ... regime transition." sounds strange and undecided. It is well established by everyday experience and experiments that the rheological properties of (granular) snow change significantly with temperature near the freezing point. Please find a more precise and informative wording.

**P1, L8:** "... the farthest deposit consists of cold snow."

**P1, L23:** "on the flow regime"
Earlier references to this phenomenon are
Gauer, P., Lied, K. and Kristensen, K. (2008). On avalanche measurements at the Norwegian full-scale test-site Ryggfonn. *Cold Reg. Sci. Technol.* **51**, 138–155.

Sovilla, B., Kern, M., Schaer, M. (2010). Slow drag in wet avalanche flow. *J. Glaciol.* **56** (198), 587–592.

**P1, L76–86:** The logical flow of this section would be improved by moving this paragraph on observations between lines 55 and 56. To make a clear connection to what follows, in L56 one could say "... is now also recognized in modeling."

**P2, L1:** In my view, calling the velocity-independent part of the impact pressure on obstacles *hydrostatic* is an unfortunate choice. Hydrostatic pressure is the pressure exerted by a fluid at rest, and the term "pressure" is commonly reserved for the isotropic part of the total stress. In the present case, there is no isotropy. Furthermore, the pressure drops significantly (but not to 0) once the avalanche has come to rest. The reason for the height dependence of the normal stress at impact is that the frictional forces between snow particles are proportional to the slope-normal stress, which is essentially of hydrostatic origin. It might be useful to borrow expressions from granular-flow mechanics and replace "dynamic" by "grain-inertia induced", "hydrostatic" by "quasi-static granular" or something similar.

**P2, L14:** "a halt"

**P2, L18:** "... and parts of which undergo a transition to a warm-wet regime"

**P2, L20:** "full" → "entire"

**P2, L21–22:** "... all the avalanching snow becomes warm and the final runout is determined by the dynamical properties of the warm flow regime."

**P2, L30:** "more slowly"

**P2, L32:** Some pictures and descriptions can be found, e.g., on the webpages http://snf.ngi.no/breitzug.040113.html and http://snf.ngi.no/breitzug.050212.html.

**P3, L2:** "lof of attention"

**P5, Fig. 1:** It seems that this figure will occupy most of an entire page, thus there is no need to squeeze things to the point where they become unintelligible. A good solution might be to give the upper panels a common main heading "Avalanche VdlS #17-3030" and each of them a subheading such as "Snowcover temperature from Alpine3D" and "Snow-temperature profiles along centerline" or similar, and analogously for the lower two panels. It took me a long time to (probably) understand the intended meaning of "$\bar{T}$ of profiles $\bar{h}$ 0.5 m depth".

**P6, L7:** I do not think you mean to say that temperature profiles cannot be measured automatically, but I cannot guess what you mean to say.

**P6, L19:** "... the typical volume of large avalanches in VdlS, $(0.5\text{--}1) \times 10^6 \, \text{m}^3$, by the typical affected are of ..."

**P6, L30:** "... crosses the threshold ..."

**P6, Eq. (2):** $H_b \rightarrow H_3$

**P7, L1–2:** What kind of "standard deviation" is meant? What kind of "law of error propagation" do you apply? From the right panels of Fig. 1 it appears that you assume fluctuations of $\bar{T}_2$ and $\bar{T}_3$ (whatever may be their origin) to be maximally correlated. If one assumed them to be maximally anti-correlated, the grey areas would become much wider at $-1°$C.

**P7, L4:** "are" $\rightarrow$ "is"

**P7, L7:** "... domain is sliced into ..."

**P7, L12 ff.:** This is an importnat passage, please describe this in somewhat more detail.

**P7, L13:** "temperature $\bar{T}$"

**P7, L17:** "at the station VDS2. The event #13-3019 . . . "

**P7, L19–20:** ". . . reflect the pattern of warm and cold temperatures reasonably well"

**P7, L25:** "an approximate" → " a minimum"

**P7, L28:** ". . . (VDS3) first became operational . . . "

**P8, Table 1:** The asterisk in "(McElwaine\* et al., 2017)" should be removed in the table legend. Also on P19, L25.
the column 'GEODAR timestamp' is difficult to read. Please use ISO notation YYYY-mm-ddTHH:MM, with the letter T separating date and time.

**P8, L2:** "such as photographs and data from the flow . . . "

**P8, L2:** I have never encountered the notion "terrain registration procedure", and a search in Google does not immediately turn up useful results. Please explain what you mean or use an established notion.

**P8, L3:** "thought of as a transfer"

**P8, L8:** Scatterbtains like me have already forgotten that this abbreviation was defined and last used only four pages ago. . .

**P8, L9:** The term 'starving–stopping mechanism' was not introduced literally before, but the readers will probably guess that you mean the same mechanism as referred to on P2, L12.

**P9, L5–6:** ". . . in the photographs in Fig. 2."

**P9, L16:** "themselves"

**P9, L18:** "extent"

**P9, L19:** Too sloppy language – a flow regime is not an area or a deposit.
"with the same colors"

**P9, L21:** "sort of lateral resolution" – please formulate this more precisely and in non-colloquial English.

**P9, L22:** "When the most distal deposits are cold, . . . "

**P9, L23–25:** Do you think that starvation is necessary in this case, or could it be enough that the front picks up warm snow and experiences higher friction? Then it would be possible for the tail to run up on the body and front.
Do you mean to say that it is (theoretically?) obvious that flows in the warm regime are slower than those in the cold regime, or do you refer to GEODAR measurements? It might be best to remove this sentence. If you keep it, you may want to write something like "The flow velocity differs markedly between the cold and warm flow regimes."

**P10, Fig. 2:** The insets lack axis labels.

**P11, Fig. 3:** The plots from the Doppler radar on the right-hand side raise a question at closer examination: The length of the range gates is 25 m in the line of sight according to information given in Sec. 2. This may correspond to about 30 m along the flow direction. At a dominant velocity of approx. 30 m/s, the front should take about 1 s to cross the range gate, which is compatible with the plots. However, between $t = 12.5$ s and 15 s, a bi-modal velocity distribution with dominant velocities diminishing from 10 to 1 m/s in the first surge and from 20 to 15 m/s in the second.

**P11, L22,26:** "farthest"

**P11, L1:** "farthest"

**P11, L10–11:** Suddenly, there seem to be different warm flow regimes. Do you perhaps mean different parts of the avalanche that are in the warm flow regime?

**P12, L7:** "decline_"

**P12, L8–9:** "... changed rather gradually ..." – the dominant velocity inside the range gate diminishes at a rate of up to 0.8 $g$, akin to an emergency stop with a car!
I cannot see velocities as high as 30  m/s in the second surge in Fig. 3 except right when it enters range gate 18.
"..., the velocity distribution ranges from ... to ..."

**P12, L15:** "from the left-hand side"

**P12, L16:** "influenced" → "wetted"

**P12, L17:** Strange sentence – how can an "altitude $H_s$" ... "visually summarize" a "snow cover"???

**P12, L22:** "discrepancy" → "difference"

**P12, L29:** "farthest"

**P13, L7:** Should one perhaps mention explicitly that this transition point is *not* a material point but more akin to a shock front?

**P13, L8:** An alternative explanation would probably be that material flowing into the range gate later is significantly slower and therefore stops more easily. In order to decide this (rather relevant) question about piling up, one would have to approximately reconstruct the flow of avalanche parcels across the range gate, adjusting the longitudinal profile of velocity so as to reproduce the recorded intensity distribution in the velocity–time plot.

**P13, L14 and throughout rest of the text:** It is never mentioned that $F_t$ is used to multiply some other physically relevant quantity. Therefore, it should be called a transition *index* rather than a transition *factor*.

**P14,15:** Figures 5 and 6 are somewhat large in this manuscript. Shrinking them by about 50% so that they can be placed side by side on a single page might still be sufficient.

**P14, L3–4:** "... have _ transition factor$\underline{s}$ $F_t = -0.18$ (...) and $F_t = +0.31$ _ (#13-3019)."

**P14, L6:** To be pedantic, one ought to say something like "..., and the set of values is well distributed over this range".

**P15, Fig. 6:** "Altitude of transition$\underline{,}$ $H_t$, against altitude of the $-1°$C _ line$\underline{,}$ $H_s$."
"happen" → "occur$\underline{s}$", "characterize$\underline{s}$"

**P15, L1–2:** If one thinks about the dynamics of avalanches and the topography of Vallée de la Sionne, it is obvious why the naïve extrapolation fails, but it would probably be helpful to some readers if this was explained.

**P16, L1:** "1700" → "1800"???

**P16, L9:** "superficial" → "surficial"

**P16, L10:** Do you mean to say that the concept of $H_s$ can be applied only to Vallée de la Sionne? What is then the value of your approach ?

**P16, L16:** "ldots of deeper_ and therefore warmer layer$\underline{s}$

**P16, L20:** "results in" → "undergoes"

**P16, L27:** "structures" → "texture",
"show_ that"
"can $\underline{produce}$ completely$\underline{}$ warm-wet deposits"

**P16, L31:** "higher" → "more"

**P17, L5:** "altitude at the end"

**P17, L5:** "gently inclined runout area"

**P17, L12–13:** "... able to hold back mass from ..." – This can easily be (mis-)read as you suggesting that tension forces are exerted on the front by the tail.

**P17, L15:** "more importantly"

**P17, L27:** "may play a role in"

**P17, L27:** "with regard to"

**P17, L34–35:** This assumption does not really affect what you have done because you have not really considered the rheology and mechanics of flowing snow. This will, however, become important when one tries to take this approach from an empirical method tied to a specific site to a general one, applicable to any avalanche path.

**P18, L15:** "relatively gentle"

**P18, L19:** "As an example,"

**P18, L23:** "common in large"

**P18, L24:** "occurs", "... of snow grains causes ..."

**P18, L28:** "unexpectedly long"

**P19, L3:** "almost all large"

**P19, L7:** "... as a first step in developing a method for predicting the..."

**P19, L22:** "... mitigation procedures be adapted..."

**Recommendation**

This paper contains a number of novel aspects, in particular the combination of several advanced experimental techniques in avalanche dynamics with snow-cover modeling. The topic of thermal effects in avalanche dynamics has recently attracted much interest, thus the paper is undoubtedly timely. The concepts discussed here may also help guiding future modeling efforts.

The data presented in this paper is unique due to the GEODAR. As far as this can be judged from the outside, the data analysis and the snow cover simulations appear to be sound.

The authors have—presumably deliberately—adopted a phenomenological approach and not tried to interpret their data through semi-quantitative or quantitative models. I personally think that a simple physical analysis, e.g. an estimate of the different components of the energy balances of the considered avalanche events along their paths, would be highly interesting and add value to the paper. Such an analysis might give some indications as to the predictive power of the transition index for avalanches in the Vallée de la Sionne and how it might be used as a predictive tool in other avalanche paths. I do not, however, insist on this point.

As mentioned under "General comments", I think it is a real pity that the authors do not present at least a preliminary analysis of the dynamical aspects that the GEODAR images put right under the readers' noses. Adding such an analysis would significantly increase the value of this paper.

Contrary to Reviewer #1, I do not have objections against the organization of the paper. I find the presentation to be logical and (except for some details mentioned above) easy to understand. The figures are informative and well executed; some minor modifications have been indicated above. However, the writing style, grammar and spelling

definitely need attention to the details.

All aspects considered, I recommend the paper for publication in The Cryosphere after minor corrections (and the mentioned additions).

---

## Author Comment (AC1) · 15 Sep 2018

**Answers to reviewer on paper:**
**"Cold-to-warm flow regime transition in snow avalanches"**

Anselm Köhler et al.

September 18, 2018

**1 Editors comments to the Authors by Guillaume Chambon**

Thank you for your submission to TC/TCD. As you may know, papers accepted for TCD appear immediately on the web for comment and review. Before publication in TCD, all papers undergo a rapid access review undertaken by the editor and/or reviewer with the aim of providing initial quality control. It is not a full review and the key concerns are fit to the journal remit, basic quality issues and sufficient significance, originality and/or novelty to warrant publication. The criteria for this evaluation can be found at `http://www.the-cryosphere.net/review/ms_evaluation_criteria.html`. Grades are from 1-4 (excellent-poor).

**Originality (Novelty): 2.**
   The study is based on methods and datasets that have already been published in several recent papers. However, the analysis proposed here goes one step beyond, focusing on avalanches that show a transition from a cold to warm flow regime, and looking for relations between the characteristics of this transition and snow cover properties. Particularly interesting is the correlation found between the degree of transition (partial or complete) and the altitude of the $-1\,°\text{C}$ isotherm in surface snow, suggesting that the transition is mainly driven by entrainment of warm snow by the avalanche.

   *Thanks for the given value of our publication, however, we would like to point out that only parts of the data have been published before and most of the applied methods are new.*

**Scientific Quality (Rigour): 2**
   The study makes clever use of a combination of advanced monitoring and modelling methods (GEODAR, snowpack modelling) to derive important and novel insight on the physics of avalanches. In that respect the paper is remarkable. I would nevertheless recommend that the authors consider expanding upon the description and analyses of their results, in order to provide better support to their conclusions. Hence, while the physical description of the transition is essentially based on two examples, I would suggest they give more information on how the features observed on these examples can be generalized, and on the variability observed among the different avalanches.

E.g., are complete cold-to-warm transitions systematically as abrupt as for the example shown in the paper?

   *Yes, we have added this observation to section 3.2.*

Along the same line, the precise definition and extraction method of path lengths $P_c$, $P_w$ and $P_t$, would need to be explained in more details.

   *We have reworked the data section which describes the extraction and terrain mapping method also based on the comment of reviewer 2.*

For complete transitions, is it always possible to measure a difference between $P_c$ and $P_t$? Is the stopping signature of the cold front always present?

   *Yes, by definition is the cold stopping signature needed otherwise we can not be sure that its a cold to warm transition or a warm shear to warm plug transition. Following that, $P_c$ is always farther than the transition point.*

For partial transitions, how is $P_w$ defined in cases where several warm fronts are visible (case of the example shown in the paper)?

   *The case of the example is quite unique, but therefor enables to study the formation of a warm tail in an insulated manner. However, much more usual is the tail above. Sometimes we can identify two warm tails, but these are each in one of the two couloirs. In this case we define it as the farthest of the two.*

I do not consider these amendments as necessary at this stage, and I will send to paper to referees in its current version, but I feel that they would contribute to render the study more convincing and sound.

**Significance (Impact): 1.**

The issue of cold to warm (or dry to wet) transitions in avalanches is emerging and of high importance, particularly in the context of climate change. Although more and more evidences show that this transition frequently plays an important role for large avalanches that travel large distances, it remains poorly documented and only seldom accounted for in models. In that respect, the paper provides important constraints to guide future developments of avalanche models.

**Presentation Quality: 3.**

I recommend that, in the revision phase, the authors consider improving the presentation of the introduction, with the objective to render the paper more self-contained. Important elements from previous studies, which are essential for proper understanding of the previous paper, would need to be recalled in more details, and the terminology would need to be made clearer. A few examples below:

- What do the authors call flow regime? How are these regimes characterized on GEODAR data? The information given at the beginning of section 3 could be usefully recalled earlier.
  We have expanded the discussion of flow regimes in the introduction, and partly merged from the beginning of section 3.

- What is called a powder snow avalanche in the paper? The fact that these avalanches involve different regimes, including a dense core, would need to be explained more clearly.
  In fact, we refer to the powder snow avalanche described by Sovilla, 2015 and Issler, 2003 with different layers and compoments. However, our data (together with Koehler 2018) shows that there are not only cold parts but also warm flow regimes involved. New to the definition is that this is the case for most of the powder snow avalanches in VdlS.

- What is the avalanche flowing length measured on radar data?
  We have added a few sentences in the methodology section to explain better how to read the MTI plots.

The clarity of the writing would also need to improved in some sections, and notation inconsistencies need to be corrected ($R_{c,w,t}$ in lieu of $P_{c,w,t}$)

During the review process, we have reworked several parts as well as the mentioned inconsitency with the notation.

**2 Anonymous Referee #1**

The paper by Köhler et al presents detailed measurements obtained from radar techniques (the GEODAR, already presented in a number of previous papers, and the pulse-Doppler system), conducted at Valle de la Sionne avalanche test site, Switzerland. The authors pay attention to the cold-to-warm transition during flow propagation, by considering in parallel the temperature calculated from the snow cover model SNOWPACK. The introduction provides some general ideas on the problem of wet-snow avalanche and more particularly on the cold-to-warm transition problem with some relevant recently published papers. Section 2 presents a brief description of the two radar techniques used and the snow cover reconstruction with SNOWPACK, as well as the data sets. Section 3 describes the results by distinguishing between complete and partial warm-to-cold transitions. It is first based on one example for each transition and two key graphs (Figs. 5 and 6) including all the avalanche events considered are then presented and shortly discussed. Section 4 is a more extended discussion on the results. The limitations of the study are discussed too. Finally, section 5 concludes the manuscript by synthesizing the main results and discussing future works to be done.

**2.1 General comment:**

The present paper relies on radar techniques that are advanced in situ measurements methods to "look inside the avalanches", following a number of recent studies (about GEODAR in particular). By coupling those radar measurements with snow cover reconstruction (the temperature in particular) and making use of the rough assumption that the temperature of the flowing snow is equal to the snow cover temperature, the authors are able to highlight a relation between the degree of cold-to-warm transition (partial versus complete) and the altitude where the snow cover temperature is $-1\,°\mathrm{C}$. Though $-1\,°\mathrm{C}$ has been previously identified as a threshold temperature controlling the transition between nearly no granulation and efficient snow granulation (see the controlled experiments done by Steinkogler et al, 2015), it may appear as an arbitrary threshold.

I enjoyed the reading of the paper. The result shown on Fig. 5 is quite remarkable. The topic addressed in the manuscript is timely. I believe that the manuscript can deserve publication if the authors make an effort to revise some points. The success of Fig. 6 is somewhat counter-balanced by the result shown on Fig. 6. My main concern on the scientific content is that a sensitivity analysis to the choice of some thresholds (thickness of $0.5\,\mathrm{m}$ for the snow cover taken into account, temperature threshold of $-1\,°\mathrm{C}$) is missing. Including such a sensitivity analysis to changing those thresholds is needed in order to reinforce the arguments provided by the authors on the physics of the cold-to-warm transition. How the plots shown on Fig. 5 and 6 would be changed by choosing other values of those thresholds?

> We have done a quick sensitivity analysis before, and also written about the results in the discussion section. The main outcome is that the found relation just shifts for different temperatures and averaging depth.
> We do not strictly propose that a linear relation between $H_s$ and the transition index is the "right answer" - we just show that there is a trend that makes sense and in the words of the second reviewer, we do not aim to provide a profound relation but rather "... (ii) to quantify in a simple way how the degree of transition from a dry-snow flow regime to a wet-snow flow regime depends on the snow- cover temperature along the avalanche path.", "They find a clear positive correlation between $F_t$ and $H_s$.".
> A detailed sensitivity study (changing 50cm and $-1\,°\mathrm{C}$ threshold) would not change anything about this trend and would not change the general message which we want to transport.

I would have a request on the organization of the paper, in addition. The discussion section (section 4) is not well-organized. I invite the authors to make it much more readable. A couple of points

that are direct interpretations of key plots shown on Figs. 5 and 6 should be discussed in more detail and moved to section 3.3. The discussion section could be split into two sub-sections: one for a general discussion on the main results and one about the limitations of the methods used.

*Thanks for the suggestions. Accordingly, we split the discussion into discussion of results and limitations of methodology.*

Please find below a detailed list of major to more minor comments on the manuscript.

**2.2 List of major/minor points:**

**Sec. 1 - Introduction**
- page 1, line 22: [..., whereas dense flow regimes, especially warm regimes, can be diverted or even stopped.]. This sentence is somewhat reductive. I agree that rapid dry- and cold-snow avalanches are difficult to divert and stop. But some flow regimes of wet-snow avalanches can pose serious problems too. Their interaction with protection structures is sometimes very complex, due to nearly unpredictable flow trajectories around avalanche dams (see some examples in Johannesson et al., 2009; European Handbook, chapter 8). Could you please qualify your statement.

*Yes, sure. As this is just a teaser sentence, we softened the statement.*

- p. 2, lines 9-10: [when liquid water is still expected to be absent.]. I would remove this statement given the fact that it is now well-established that localized melting can occur at ambient temperatures a few degrees below the freezing point (Dash et al, RMP 2006; Turnbull, PRL 2011).

*Removed the statement as suggested.*

- p. 2, end of line 10: the existence of that quasi-liquid layer in flowing snow has two consequences. It can increase snow cohesion on the one side (and thus increase the size of aggregates) but it may also lubricate the contacts between snow aggregates on the other side (thus enhance flow mobility). Maybe the second effect could be shortly discussed, in addition.

*We have added a sentence about lubrication and slush flows.*

- p. 2, lines 20-21: [A partial transition affects only the tail of the flowing avalanche and the final run-out is still cold-dominated.]. This sentence suggests that the transition does not occur at the front but mostly at the tail. Could it be that such a scenario with the cold-to-warm transition occurring at the front does exist?

*We have reformulated the sentence, stating that partial transition means that warm and cold regimes separate due to different velocities and thus, the warm regime becomes visible often only at the tail. In our opinion, a transition may also occur at the front if for example the lower layers are warmer then the upper layers; but there is no way to resolve that with GEODAR.*

- p. 3, lines 13-14: why this arbitrary value of 0.5 m? The statement [Despite the crudeness of this measure, we assume that...] needs clarification. Please could you justify? Maybe you could explicitly refer to the paper section which addresses in detail the assumptions made.

*We have connected this statement with the justification of the 0.5 m assumption in section 2.2.*

- p. 3, lines 20-21: [Finally, the discussion (sec. 4) brings together both result parts and the study is finished by a conclusion (sec. 5).]. This sentence is quite easy: I guess you could propose a more precise sentence, more relevant to the content of your paper in order to announce both discussion and conclusion sections.

    Yes, we have reformulated the sentences to introduce the content.

**Sec. 2 - Methods and data**

  - p. 4, line 15-16: maybe you could give (at least) one relevant reference already published for each system, the older system and the newer system.

    Ok.

- Fig. 1, caption, line 3 (p. 5): shown (not show)

    Ok.

- p. 6, lines 12-23: this part justifies your assumptions made (in particular the 0.5 m). Please refer to this part at p. 3, line 13, in the introduction (see a previous comment).

    Ok.

- p. 8, Table 1: could you please provide an order of magnitude of the error/uncertainty on $P_c$ and $P_w$? And thus $F_t$?

    We state an uncertainty for $P_c$ and $P_w$ in the data section 2.3 already. We have added a sentence on the error in $F_t$ where we define it in section 3.3. The error is in the order of 0.05 to 0.1.

- p. 9, lines 10-13: is there any uncertainty on this threshold of $-1\,°C$ between warm and cold regimes? Temperature is certainly a very important control parameter but other factors may come into play. Maybe you could discuss this a bit (see another comment below, on Fig. 6).

    Sure, there will be an uncertainty on the $-1\,°C$ threshold. However, the data in this paper can not be used to validate or test this threshold and we simply take it from the literature (see introduction). We think that much more difficult to handle is the assumption that the flowing snow temperature is assumed to be similar to the resting snow cover temperature. This assumption is probably one of the reasons explaining the scatter in Fig. 6. We have added a paragraph about that.

**Sec.3 - Results**

  - p. 12, lines 3-12: this is a very interesting observation, providing a quantitative proof of a mechanism known from the field experience gained by some snow avalanche experts. Under a context of climate change / global warming, we may expect more events with rain occurring at high altitude on the snow cover during winter. Your measurements are relevant to this problem. Maybe you could add a short word on this point here.

    Yes, there is some work done by C. Mitterer et al. about the snow cover weakening due to melting in spring time and they successfully relate the beginning of a wet avalanche period to the liquid water content (LWC). That such increase in LWC can come from rain is just an interesting observation but a bit off-topic to the rest of the content.
    Actually this paper is practically related to the climate change problematic. Transitional avalanche are exactly what we expect to happen more in future. This can be also independent of rain. Transition will happen more in future just because we expect the snow cover to increase in temperature. A final paragraph in the conclusion states that.

- p. 12, lines 21-24: [... This discrepancy corroborates the turbulent character of both surges.]. Could you please explain better what you mean here? Do the differences stem from different positions of the devices and/or assumptions made with respect to main flow direction? As such, very turbulent flows, with significant velocities in all (3D) directions can produce different results depending on the technique used. This part needs more clarification.

> No, both devices are mounted at the same location. Only the Doppler radar measures velocity directly and it's the full velocity distribution of the complete frontal zone. Instead, GEODAR measures only the front approach velocity. Large differences between the approach velocity (GEODAR) and the Doppler velocity distribution are usually found in the intermittent regime of powder snow avalanches, where large mesoscale structures can have velocities up to 60% larger than the front. Since these structures are measured by the Doppler radar, the measured velocities are generally bigger. Characteristic for the intermittent regime is surging, which originates from a non-uniform velocity field, see Koehler 2016 and Fischer 2016.

- p. 13, Eq. (5): could you please give an uncertainty on $F_t$? (back to a previous comment on uncertainties on $P_w$ and $P_c$). And report this uncertainty on Fig. 5.

> We added a paragraph on the uncertainty of $F_t$ right after the definition in equation 5. Given that $P_{w,c}$ can have an error of 100 m, we find $F_t$ to have an error of up to 0.1.

- Fig. 5: it is nice to see this correlation between $H_s$ and your $F_t$. Would be nice too to study the sensitivity of the plot to changing the threshold of $-1\,°C$. Would that plot be improved or deteriorated by choosing a different temperature threshold (below or above $-1\,°C$)?

> No, the plot works exactly the same, just $H_s$ is shifted. We have tested that by choosing different temperature thresholds. But we find that $-1\,°C$ is a reasonable choice: The limit towards $F_t = 1$ (pure warm avalanches) is in the release area and not above or below. Partial and complete transitions are split by the $H_s:H_t$ 1:1-line (Figure 6). We have added a paragraph in the discussion of the methodology.

- p. 14 - 15. That you use the linear fit to extrapolate and obtain the value of 860 m a.s.l. for $F_t$=-1 is questionable to me. Because it does concern the arrest conditions of the avalanche, I guess the effect of local topography coupled with the snow (flowing/deposited and entrained) properties is crucial. I would suggest that either you dont extrapolate or your provide more critical discussion on that result.

> Well, we do not extrapolate explicitly. We just discuss the linear relation and state already that it does not work towards $F_t$=-1. However, we highlighted the limited validity towards $F_t$=-1 more clearly.

- p. 15, lines 3-6, and Fig. 6: I may interpret this plot showing $H_s$ versus $H_t$ as a proof that (i) the $-1\,°C$ threshold may a bit arbitrary and (ii) other factors come into play. Those points need more critical discussion. Maybe some arguments given in the discussion should be already developed here (see another comment thereafter).

> This is correct. We do discuss all this in the long discussion section. For example, we state that our choice of temperature threshold of $-1\,°C$ splits the partial and complete transitions. However, the main point taken from this plot is that we need to differentiate between superficial and deep layer entrainment.

**Sec.4 - Discussion**
- p. 16, lines 10-11: that the flow regime in the run-out zone can be estimated when $H_s$ is known relies on the linear fit proposed for the relation between $F_t$ and $H_s$. You could be more precise here, and add at the same time that this will need further investigation: linear fit or other relation? range of $F_t$ for which the linear fit is valid? asymptotic behaviors when $F_t$ tends towards -1 or +1?. See also a previous comment.

> We added a paragraph which explicitly includes the limitations and the necessity of further investigations. This paragraph is similar to what the second reviewer D. Issler suggested with the three bullet points.

- a general comment: this section is difficult to read because there are too many ideas. I would propose to put some points (in particular: entrainment at the surface versus deeper in the snow cover, effect of the topography, front dynamics) earlier in Sec. 3 and maybe extend the discussion on those points in Sec. 3, because they are direct and important interpretations of the plots shown on Figs. 5 and 6. Also, the remaining points (not transferred to sec. 3) could be a sub-section 4.1 and the discussion on the limitations of the method (starting from line 26, p. 17) could be a sub-section 4.2.

> We have split the discussion into results and limitations.

**Sec. 5 - conclusions**
  - p. 18, line 26: the flow regime influences not only the pressure on structures but also the flow mobility (run-out: velocity and volume). Please add those points.

> Thanks, we have added flow mobility.

- p. 18, line 32: please remove "robust relation" but (for instance) use "correlation" instead or keep "relation" only. I agree that this result is very nice but this result will need further validation.

> Removed robust.

- p. 19, lines 3-6: how those values of 300 m and 500 m depend on some (arbitrary) choices you made? A sensitivity analysis (of plots on Fig. 5 and Fig. 6) to changing the threshold values for the temperature ($-1\,°C$ here) and the snow cover thickness taken into account ($0.5\,m$) is missing in your study.

> Sure, these values are found with our assumptions of $0.5\,m$ surface snow and $-1\,°C$ threshold temperature. Other factors like flow volume, path geometry, older deposits ... are not taken into account, but undoubtedly important. Any future study needs to investigate those factors in order to find a universal relation. We do not believe that a sensitivity study of the above mentioned values brings any more insights, rather than shifting the relation. We have added that these values are found by taking the chosen values for the threshold temperature and the depth.

**3  D. Issler (Referee #2)**

**3.1  Content of the paper**

Over a period of some three decades, our capability of numerically simulating the evolution of the snow cover in some detail in 3D has developed to a level where these tools can be used in diverse applications with some confidence (notwithstanding major residual problems). Greatly developed and diversified experimental techniques – all of them installed at the Valle de la Sionne test site in Switzerland – have given us an unprecedented, detailed view of the processes inside flowing avalanches. One of the most conspicuous of these new instruments is GEODAR, a phase-array-based radar system that eventually will allow high-resolution 3D mapping of entire avalanches through time. Finally, thanks to IR imaging, the role of the snow temperature in the snow cover and the flowing avalanche has become a major focal point of research in avalanche science, mostly due to work at SLF in Switzerland.

In this paper, the authors combine these three major elements: snow-cover simulation, 18 avalanche measurements with GEODAR and Doppler radar, as well as previously gained insight into the critical role of snow temperatures near melting on the flow regime of avalanches. The main objectives are (i) to verify the effect of snow-cover temperature by comparing a rather large sample of measured avalanches and (ii) to quantify in a simple way how the degree of transition from a dry-snow flow regime to a wet-snow flow regime depends on the snow-cover temperature along the avalanche path.

The measurements with GEODAR and Doppler radar allow distinction between different flow regimes, as recently shown in a different paper by some of the same authors. Moreover, they can distinguish "complete" from "partial" transitions and locate a representative transition point. From the run-out distances of the warm and cold parts of the flow, they construct a transition index $F_t$ and relate it, for each event, to the reconstructed altitude $H_s$ where the mean temperature of the top layers of the snow cover reached $-1\,°C$. They find a clear positive correlation between $F_t$ and $H_s$. Similarly, warm avalanches (undergoing a full transition) are shown to make the transition to the warm-snow flow regime above NO: BELOW the altitude $H_s$, whereas that point is below NO: ABOVE $H_s$ for all cold-snow avalanches (with partial transition only).

Thank you for summarizing the paper in such good words. We even allowed us to copy some sentences into the discussion and conclusion.

**3.2  General comments**

The transition index proposed by the authors is a clever attempt to (semi-)quantitatively capture an aspect of the flow-regime transition process with a minimum number of observable quantities. In order to link it to the thermal regime of the flow, the transition altitude, $H_t$, is invoked and statistically compared to the altitude $H_s$ below which the upper snow cover is warm. The authors seem to be aware of the difficulties and limitations of this approach, but it might be useful to spell them out more explicitly. From my point of view, the following points are particularly important:

- The transition index will probably be most useful for avalanches with drop heights of 500 m or more. For smaller avalanches, $H_s$ tends to be either above the release area or below the run-out area.

- While $H_s$ can be determined wherever and whenever there is enough meteorological data for running snow-cover simulations, finding $H_t$ for a given event requires either detailed investigation of the avalanche deposits or measurements with a GEODAR or advanced Doppler radar.

- For use as a predictive tool, e.g. for road closures or evacuations, a plot like Fig. 6, containing many events, would be necessary. Probably, such copious and detailed data is available only for a handful of avalanche paths worldwide.

We adopted your suggestion to highlight these limitations with bullet points, and placed the list at the beginning of the discussion to bring these important points directly to the reader.

That being said, I agree, however, that the transition index and the correlation between $H_s$ and $H_t$ are a meaningful way of demonstrating the relevance of the thermal regime for the flow of avalanches.

The method for determining the uncertainty in the snow temperatures $\bar{T}_{2,3}$ remains unclear to me. The way I read the text, they calculate the standard deviations as

$$\sigma_T = [\sum_{1}^{N}(T_i - \bar{T})^2/(N-1)]^{1/2},$$

with the sum extending over the computational layers used by SNOWPACK in the top $0.5\,\mathrm{m}$ of the snow cover. If this is indeed what they mean, I cannot see how this should be connected to the uncertainty of $H_s$ – that uncertainty is more directly connected to the question whether a linear extrapolation of snow pack temperatures is admissible. As a consequence, I cannot assess whether the authors approach for determining the consequent uncertainty in $H_s$ is sound or too optimistic. The way they do it according to Fig. 1 assumes that the deviations of $T_2$ and $T_3$ from their means $\bar{T}_2$ and $\bar{T}_2$ are tightly and positively correlated. If this is not the case, the uncertainty in $H_s$ will be much larger. This would, however, have considerable importance for Figs. 5 and 6 and for the firmness of conclusions that can be drawn from them. These issues have also been commented upon by the other reviewer and need to be addressed carefully by the authors.

Yes, that's what we did. And in our opinion this is one of the few suitable and possible approaches how we can estimate an uncertainty of the snow cover temperature. We mostly think of daily variation of the temperature which propagates into the snow cover slowly (warming as soon as the sun comes out). Such warming and in particular the timing may not be captured by the flat field measurements and their simulations, but the spread of layer temperatures may help to get a feeling of the resulting uncertainty or spread of possible values.
Since it is not a standard deviation or error estimate – as you clearly pointed out – we changed the name to temperature variability.
Given the fact that the "error" is the variability of the snow temperatures in the top 50cm of the snow cover, we think that the question of correlated or anti-correlated error between both weather stations comes down to fluctuations of the SNOWPACK model result. Therefore, one can expect that the variations rather go into the same direction for both weather stations. The linear extrapolation of the snow temperatures is in fact motivated by the work of Steinkogler et al. (2014, crst). They find a close to linear temperature distribution with 4 points along the VDLS avalanche path (taken from Alpine3D simulations). Here, we did use Alpine3D for all avalanches to check the linearity, and as we state, the examples in Figure 1 are the ones with the largest deviation from linear.

The title of the paper is more general than its content in that dynamical aspects are more or less completely left out. However, the GEODAR data offer a unique opportunity to quantify some aspects of the dynamics: Figures 3 and 4 suggest that a major component of the avalanche first moves at a nearly constant speed, then decelerates over a period of 5 - 10 s, and then continues again at nearly constant speed. From the curvature of the streaks, it should be relatively straightforward to extract the deceleration, and since the location is also known fairly precisely, the retarding accelerations before, during and after the cold-to-warm flow-regime transition can be determined.

This is a rather remarkable phenomenon with far-reaching consequences for modeling the flow. I do not understand why the authors hardly mention this, and I strongly encourage them to dedicate a subsection or a few paragraphs to an at least preliminary analysis.

> Yes, you are completely right: It is a pity that we left out those dynamic aspects. However, an analysis like this can easily fill a paper itself: Yes, GEODAR suggests to show all these features directly, but the problem is more subtle and needs a lot of care!
> Are we sure, that the snow is the same before/after the transition, but just a bit warmer? Where exactly are the plug flow regime parts in the avalanche, do they have a down-ward or side-way direction? Similarly, are the features at the same lateral position or do they belong to other features of the avalanche?
> To answer those questions precisely, one needs a very complete reported dataset – For both examples we do not even have a video because of cloudy weather and night. In this respect, we will not dynamically investigate the dataset here, but we are planing to find out scientifically sound methods to do that in future.

**3.3 Minor points**

P1, L4,5: The sentence "The intake of ... regime transition." sounds strange and undecided. It is well established by everyday experience and experiments that the rheological properties of (granular) snow change significantly with temperature near the freezing point. Please find a more precise and informative wording.

> Changed the sentence to point out the importance of warm snow entrainment on cold to warm flow regime transitions.

P1, L8: "... the farthest deposit consists of cold snow."

> Ok.

P1, L23: "on the flow regime" Earlier references to this phenomenon are
Gauer, P., Lied, K. and Kristensen, K. (2008). On avalanche measurements at the Norwegian full-scale test-site Ryggfonn. Cold Reg. Sci. Technol. 51, 138- 155.

Sovilla, B., Kern, M., Schaer, M. (2010). Slow drag in wet avalanche flow. J. Glaciol. 56 (198), 587-592.

> Thanks for the literature suggestion. We have added Gauer (2008), but keep Sovilla (2016) as it compares the pressure for different flow regimes rather than solely analyzing wet snow avalanches.

P1, L76 – 86: The logical flow of this section would be improved by moving this paragraph on observations between lines 55 and 56. To make a clear connection to what follows, in L56 one could say "... is now also recognized in modeling."

> We are sorry, but we can not identify the paragraph you are suggesting to move. There are no line numbers larger than 24 on P1.

P2, L1: In my view, calling the velocity-independent part of the impact pressure on obstacles hydrostatic is an unfortunate choice. Hydrostatic pressure is the pressure exerted by a fluid at rest, and the term "pressure" is commonly reserved for the isotropic part of the total stress. In the present case, there is no isotropy. Furthermore, the pressure drops significantly (but not to 0) once the avalanche has come to rest. The reason for the height dependence of the normal stress at impact is that the frictional forces between snow particles are proportional to the slope-normal stress, which is essentially of hydrostatic origin. It might be useful to borrow expressions from granular-flow mechanics and replace "dynamic" by "grain-inertia induced", "hydrostatic" by "quasi-static granular" or something similar.

Thank you for the suggested expressions. We adopted grain-inertia induced pressure and quasi-static gravitational contribution. Both terms are used in Sovilla (2016) which is the relevant literature for the sentence.

P2, L14: "a halt"

Ok.

P2, L18: "...and parts of which undergo a transition to a warm-wet regime"

Ok.

P2, L20: "full" $\Rightarrow$ "entire"

Ok.

P2, L21-22: "...all the avalanching snow becomes warm and the final runout is determined by the dynamical properties of the warm flow regime."

Ok.

P2, L30: "more slowly"

Ok.

P2, L32: Some pictures and descriptions can be found, e.g., on the webpages `http://snf.ngi.no/breitzug.040113.html` and `http://snf.ngi.no/breitzug.050212.html`.

These two links are very interesting and indeed describe events with complete transition. We would like to include them in the paper, but we think, their content should be uploaded to a data repository as websites can change too quickly. We suggest for example the European repository `zenodo.org`. Zenodo offers a DOI and so-called communities for the content which make linking very easy.

P3, L2: "lof of attention"

Ok.

P5, Fig. 1: It seems that this figure will occupy most of an entire page, thus there is no need to squeeze things to the point where they become unintelligible. A good solution might be to give the upper panels a common main heading "Avalanche VdlS #17-3030" and each of them a subheading such as "Snowcover temperature from Alpine3D" and "Snow-temperature profiles along centerline" or similar, and analogously for the lower two panels. It took me a long time to (probably) understand the intended meaning of "$\bar{T}$ of profiles $\bar{h}$ 0.5 m depth".

Thanks for the feedback, indeed this figure contains a lot of information. We have improved the organization with headings, clearer legend and caption.

P6, L7: I do not think you mean to say that temperature profiles cannot be measured automatically, but I cannot guess what you mean to say.

Changed accordingly to mean that temperature profiles are not measured automatically.

P6, L19: "...the typical volume of large avalanches in VdlS, $(0.5 - 1) \cdot 10^6 \, m^3$ , by the typical affected are of ..."

Ok.

P6, L30: "...crosses the threshold ..."

Ok.

P6, Eq. (2): $H_b \Rightarrow H_3$

Ok.

P7, L1-2: What kind of "standard deviation" is meant? What kind of "law of error propagation" do you apply? From the right panels of Fig. 1 it appears that you assume fluctuations of $\bar{T}_2$ and $\bar{T}_3$ (whatever may be their origin) to be maximally correlated. If one assumed them to be maximally anti-correlated, the grey areas would become much wider at $-1\,°\mathrm{C}$.

As said above, we use the temperature fluctuations or variations found in the simulated layers of the top 50 cm. This is used to get a range of possible value for the $H_s$, as it is one way we can estimate any uncertainty.
The question if the fluctuations are correlated or anti-correlated is complicated and hard to answer. We can only argue that since they are fluctuations, their origin is likely to be the same on both stations and thus they would correlate. The question can also be expanded in the direction of the reliability of SNOWPACK – we hardly have a "proof" that the simulation setup is correct for snow temperatures (it is often calibrated with the total snow depth so that accumulation and melting is correct on the scale of the season). We clarified the definition of the temperature uncertainty in the new version of the paper.

P7, L4: "are" $\Rightarrow$ "is"

Ok.

P7, L7: "...domain is sliced into ..."

Ok.

P7, L12 ff.: This is an important passage, please describe this in somewhat more detail.

We hope to have clarified this paragraph together with the caption of Figure 1 and the figure itself.

P7, L13: "temperature $\bar{T}$"

Ok.

P7, L17: "at the station VDS2. The event #13-3019 ..."

Ok.

P7, L19-20: "...reflect the pattern of warm and cold temperatures reasonably well"

Ok.

P7, L25: "an approximate" ⇒ " a minimum"

Ok.

P7, L28: "...(VDS3) first became operational ..."

Ok.

P8, Table 1: The asterisk in "(McElwaine* et al., 2017)" should be removed in the table legend. Also on P19, L25. the column GEODAR timestamp is difficult to read. Please use ISO notation YYYY-mm-ddTHH:MM, with the letter T separating date and time.

Changed the citation style. We would like to stay with the GEODAR timestamp, as this is the notation used in the data repository, and we need the additional seconds to uniquely identify the data sets.

P8, L2: "such as photographs and data from the flow ..."

Ok.

P8, L2: I have never encountered the notion "terrain registration procedure", and a search in Google does not immediately turn up useful results. Please explain what you mean or use an established notion.

We simplified the sentence. One could use "geo-referencing scheme", but even that is just a word for a bunch of techniques. To explain what we did, we cite Koehler (2016) which covers the process in detail.

P8, L3: "thought of as a transfer"

Ok.

P8, L8: Scatterbrains like me have already forgotten that this abbreviation was defined and last used only four pages ago...

We reformulated the sentence to indicate that MTI plot is a product of the GEODAR data and link with Fig. 2.

P8, L9: The term starving-stopping mechanism was not introduced literally before, but the readers will probably guess that you mean the same mechanism as referred to on P2, L12.

We changed the word order to make the guessing easier.

P9, L5-6: "...in the photographs in Fig. 2."

Ok.

P9, L16: "themselves"

Ok.

P9, L18: "extent"

Ok.

P9, L19: Too sloppy language – a flow regime is not an area or a deposit. "with the same colors"

Added: regions of the flow regimes.

P9, L21: "sort of lateral resolution" – please formulate this more precisely and in non-colloquial English.

We removed this sentence, as it is more confusing than helpful.

P9, L22: "When the most distal deposits are cold, . . . "

Ok.

P9, L23-25: Do you think that starvation is necessary in this case, or could it be enough that the front picks up warm snow and experiences higher friction? Then it would be possible for the tail to run up on the body and front. Do you mean to say that it is (theoretically?) obvious that flows in the warm regime are slower than those in the cold regime, or do you refer to GEODAR measurements? It might be best to remove this sentence. If you keep it, you may want to write something like "The flow velocity differs markedly between the cold and warm flow regimes."

Actually starvation is a good word, if for example the runout of the avalanche in the cold regime is reached by the intermittent region. In this case, mesoscale structures starve by depositing material with time. In our opinion, a lack of light snow available to entrain is what causes a starving in the more dilute part of the flow (which may have a limited interaction with the ground). However, if the furthest reaching part of the cold avalanche is a denser flow, there may be the intake of warmer snow that changes the flow regime. Concerning the velocity, you are right and we have removed the lines.

P10, Fig. 2: The insets lack axis labels.

Ok.

P11, Fig. 3: The plots from the Doppler radar on the right-hand side raise a question at closer examination: The length of the range gates is 25 m in the line of sight according to information given in Sec. 2. This may correspond to about 30 m along the flow direction. At a dominant velocity of approx. 30 m/s, the front should take about 1 s to cross the range gate, which is compatible with the plots. However, between t = 12.5 s and 15 s, a bi-modal velocity distribution with dominant velocities diminishing from 10 to 1 m/s in the first surge and from 20 to 15 m/s in the second.

We have a problem with this question which seems to be not finished, probably? We are not sure what the point of the reviewer was, but maybe it is important to note that due to the missing lateral resolution of the Doppler radar it may appear that surges co-exist or overtake each other which may also lead to the second surge that appears to be a bi-modal velocity distribution but in reality is a consequence of the overlapping of lateral avalanche features.

P11, L22,26: "farthest"

Ok.

P11, L1: "farthest"

Ok.

P11, L10-11: Suddenly, there seem to be different warm flow regimes. Do you perhaps mean different parts of the avalanche that are in the warm flow regime?

This is indeed confusing. We have changed flow regime to singular.

P12, L7: "decline "

Ok.

P12, L8-9: "...changed rather gradually ..." - the dominant velocity inside the range gate diminishes at a rate of up to 0.8 g, akin to an emergency stop with a car! I cannot see velocities as high as 30 m/s in the second surge in Fig. 3 except right when it enters range gate 18. "..., the velocity distribution ranges from ...to ..."

We have removed the word gradually. And reformulated the complete passage to include that we see a rapid deceleration along a single range gate for each surge, but also a deceleration between the range gates for surge 2 and 3 whereas the first front continues through all three range gates.

P12, L15: "from the left-hand side"

Ok.

P12, L16: "influenced" ⇒ "wetted"

Ok.

P12, L17: Strange sentence - how can an "altitude $H_s$ " ... "visually summarize" a "snow cover"???

P12, L22: "discrepancy" ⇒ "difference"

Ok.

P12, L29: "farthest"

Ok.

P13, L7: Should one perhaps mention explicitly that this transition point is not a material point but more akin to a shock front?

Yes, to mention explicitly that no material is travelling uphill is a good idea as the MTI plot are often quite confusing to read. We have added a sentence.

P13, L8: An alternative explanation would probably be that material flowing into the range gate later is significantly slower and therefore stops more easily. In order to decide this (rather relevant) question about piling up, one would have to approximately reconstruct the flow of avalanche parcels across the range gate, adjusting the longitudinal profile of velocity so as to reproduce the recorded intensity distribution in the velocity-time plot.

Thanks for bringing our thoughts to this possible interpretation. We have added that, but also mentioned that a decision is difficult to make. As you mention, to decide between both interpretations needs a lot of effort, and this is exactly the same for many of the "dynamical aspects put right under the readers nose".

P13, L14 and throughout rest of the text: It is never mentioned that $F_t$ is used to multiply some other physically relevant quantity. Therefore, it should be called a transition index rather than a transition factor.

Yes, we have though about that too, however, we changed it throughout the paper.

P14,15: Figures 5 and 6 are somewhat large in this manuscript. Shrinking them by about 50% so that they can be placed side by side on a single page might still be sufficient.

Yes sure, to our knowledge the final layout is in two columns so that the figures can be placed in a single cumn each. Changed the figure size to half page width in the draft.

P14, L3-4: "...have transition factors $F_t$ = -0.18 (...) and $F_t$ = +0.31 (#13-3019)."

Ok.

P14, L6: To be pedantic, one ought to say something like "..., and the set of values is well distributed over this range".

Ok.

P15, Fig. 6: "Altitude of transition, $H_t$ , against altitude of the $-1\,°C$ line, $H_s$ ." "happen" $\Rightarrow$ "occurs", "characterizes"

Ok.

P15, L1-2: If one thinks about the dynamics of avalanches and the topography of Vallée de la Sionne, it is obvious why the naive extrapolation fails, but it would probably be helpful to some readers if this was explained.

We have added a sentence describing roughly the avalanche path at the beginning of the method section.

P16, L1: "1700" $\Rightarrow$ "1800"???

Yes, we can say that and it fits with the next paragraph where we find the $-1\,°C$ line above 1800.

P16, L9: "superficial" $\Rightarrow$ "surficial"

Ok.

P16, L10: Do you mean to say that the concept of $H_s$ can be applied only to Valle de la Sionne? What is then the value of your approach ?

Well, we hope not, but we can not test it yet. We have included explicitly the unknown path dependency into the beginning of the discussion section.

P16, L16: "...of deeper and therefore warmer layers

    Ok.

P16, L20: "results in" ⇒ "undergoes"

    Ok.

P16, L27: "structures" ⇒ "texture", "show that" "can produce completely warm-wet deposits"

    Ok.

P16, L31: "higher" ⇒ "more"

    Ok.

P17, L5: "altitude at the end"

    Ok.

P17, L5: "gently inclined runout area"

    Ok.

P17, L12-13: "...able to hold back mass from ..." - This can easily be (mis-)read as you suggesting that tension forces are exerted on the front by the tail.

    Thanks for pointing out this sentence. We have clarified it.

P17, L15: "more importantly"

    Ok.

P17, L27: "may play a role in"

    Ok.

P17, L27: "with regard to"

    Suggestion not found

P17, L34-35: This assumption does not really affect what you have done because you have not really considered the rheology and mechanics of flowing snow. This will, however, become important when one tries to take this approach from an empirical method tied to a specific site to a general one, applicable to any avalanche path.

    Right, that is the beauty of an empirical approach that many factors are included. However, we left the sentence and pointed the reader that this becomes important when generalizing the results.

P18, L15: "relatively gentle"

Ok.

P18, L19: "As an example,"

Ok.

P18, L23: "common in large"

Ok.

P18, L24: "occurs", "...of snow grains causes ..."

Ok.

P18, L28: "unexpectedly long"

Ok.

P19, L3: "almost all large"

Ok.

P19, L7: "...as a first step in developing a method for predicting the..."

Ok.

P19, L22: "...mitigation procedures be adapted..."

Ok.

**3.4   Recommendation**

This paper contains a number of novel aspects, in particular the combination of several advanced experimental techniques in avalanche dynamics with snow-cover modeling. The topic of thermal effects in avalanche dynamics has recently attracted much interest, thus the paper is undoubtedly timely. The concepts discussed here may also help guiding future modeling efforts.

The data presented in this paper is unique due to the GEODAR. As far as this can be judged from the outside, the data analysis and the snow cover simulations appear to be sound.

The authors havepresumably deliberatelyadopted a phenomenological approach and not tried to interpret their data through semi-quantitative or quantitative models. I personally think that a simple physical analysis, e.g. an estimate of the different components of the energy balances of the considered avalanche events along their paths, would be highly interesting and add value to the paper. Such an analysis might give some indications as to the predictive power of the transition index for avalanches in the Valle de la Sionne and how it might be used as a predictive tool in other avalanche paths. I do not, however, insist on this point.

We agree that the analysis could be extended. However, the described avalanches are all very different and, in our opinion, any more indepth analysis would require to deal with each one individually rather than describing all with a common method. We will work on exactly that in future projects and thus build on the results shown here.

As mentioned under "General comments", I think it is a real pity that the authors do not present at least a preliminary analysis of the dynamical aspects that the GEODAR images put right under the readers noses. Adding such an analysis would significantly increase the value of this paper.

> We agree that the GEODAR data together with the Doppler radar data include many details and dynamical aspects on each avalanche. However, even very simple questions require a detailed knowledge of each event. The work definitively continues in interpreting the data in the mentioned details.

Contrary to Reviewer #1, I do not have objections against the organization of the paper. I find the presentation to be logical and (except for some details mentioned above) easy to understand. The figures are informative and well executed; some minor modifications have been indicated above. However, the writing style, grammar and spelling definitely need attention to the details.

All aspects considered, I recommend the paper for publication in The Cryosphere after minor corrections (and the mentioned additions).

---

## Author Response (AR2)

**Answers to report on review of paper:**
**"Cold-to-warm flow regime transition in snow avalanches"**

Anselm Köhler et al.

November 9, 2018

**1 Editors comments to the Authors by Guillaume Chambon**

I have now received one report on your revised manuscript. Following the positive appreciation of the reviewer, I am happy to announce that you paper is accepted for publication in the Cryosphere. Note that the reviewer pointed out a handful of technical corrections that you may wish to consider prior to submitting the final version.

Best regards, Guillaume Chambon / TC Topical Editor.

> We are pleased that the reviewed version of our manuscript now is suitable for publication. We thank the reviewers and the editor for their great work, and edited the manuscript according the suggested corrections. Furthermore, we have finalized the bibliography by adding the missing DOI to Sovilla, 2018, and added the ISBN number to UNESCO, 1981 entry. Note, we have remove all our own `newcommands` (e. g. alpine3d, snowpack and VDLS), which appear now in the track changes document a bit weird.

**2 Anonymous reviewer:**

I feel that the revised version is now suitable for publication in The Cryosphere. The authors made a substantial effort to clarify the list of issues another referee (Dieter Issler) and I raised in our initial reports and improve their manuscript accordingly.

I therefore recommend the paper to be published.

Note that I found some minor (editing) points to be considered before final publication:

-"equation 4" (page 8, line 4) and "Eq. 4" (elsewhere in the manuscript) are both used. This should be fixed according to 'The Cryosphere' editing rules.

We have adopted the abreviation of equation, sections and figures according to the journal rules.

-page 10, line 26: it looks like as if "both" is not at the right place? I would put "both" after "by means of" instead

Ok.

-page 15, line 9: should be "propagates"?

Ok.

-page 17, line 2: should be "needs"?

Ok.

-page 20, line 5: should be "occurs"?

Ok.

[revised manuscript text omitted]